# Modulation of fracture healing by the transient accumulation of senescent cells

Dominik Saul[1,2,3], David G Monroe[1,2,4], Jennifer L Rowsey[1,2], Robyn Laura Kosinsky[5], Stephanie J Vos[1,2], Madison L Doolittle[1,2], Joshua N Farr[1,2,4]*, Sundeep Khosla[1,2,3,4]*

[1]Division of Endocrinology, Mayo Clinic, Rochester, United States; [2]Robert and Arlene Kogod Center on Aging, Mayo Clinic, Rochester, United States; [3]Department of Trauma, Orthopedics and Reconstructive Surgery, Georg-August-University of Goettingen, Goettingen, Germany; [4]Division of Physiology and Biomedical Engineering, Mayo Clinic College of Medicine, Mayo Clinic, Rochester, United States; [5]Division of Gastroenterology and Hepatology, Mayo Clinic, Rochester, United States

**Abstract** Senescent cells have detrimental effects across tissues with aging but may have beneficial effects on tissue repair, specifically on skin wound healing. However, the potential role of senescent cells in fracture healing has not been defined. Here, we performed an in silico analysis of public mRNAseq data and found that senescence and senescence-associated secretory phenotype (SASP) markers increased during fracture healing. We next directly established that the expression of senescence biomarkers increased markedly during murine fracture healing. We also identified cells in the fracture callus that displayed hallmarks of senescence, including distension of satellite heterochromatin and telomeric DNA damage; the specific identity of these cells, however, requires further characterization. Then, using a genetic mouse model (*Cdkn2a*[LUC]) containing a *Cdkn2a*[Ink4a]-driven luciferase reporter, we demonstrated transient in vivo senescent cell accumulation during callus formation. Finally, we intermittently treated young adult mice following fracture with drugs that selectively eliminate senescent cells ('senolytics', Dasatinib plus Quercetin), and showed that this regimen both decreased senescence and SASP markers in the fracture callus and significantly accelerated the time course of fracture healing. Our findings thus demonstrate that senescent cells accumulate transiently in the murine fracture callus and, in contrast to the skin, their clearance does not impair but rather improves fracture healing.

*For correspondence:
Farr.Joshua@mayo.edu (JNF);
khosla.sundeep@mayo.edu (SK)

Competing interest: The authors declare that no competing interests exist.

## Introduction

The worldwide burden of fractures is growing. One reason for this major public health problem is population aging, which is associated with progressive physiological decline, leading to an increased risk for many chronic diseases (*Khosla et al., 2020*). Hallmarks of aging that drive this deterioration include genomic instability, telomere attrition, epigenetic alterations, loss of proteostasis, deregulated nutrient sensing, mitochondrial dysfunction, stem cell exhaustion, altered intercellular communication, and cellular senescence (*López-Otín et al., 2013*). Recently, cellular senescence has emerged as a promising therapeutic target to prevent aging of multiple tissues, including the musculoskeletal system (*Chandra et al., 2020*; *Farr et al., 2017*; *Farr et al., 2019*; *Farr and Khosla, 2019*; *Khosla et al., 2018*; *Khosla et al., 2020*; *Xu et al., 2018*). Because the 'geroscience hypothesis' postulates that targeting fundamental mechanisms of aging, such as cellular senescence, can delay the onset of age-associated diseases as a group, a better understanding of the beneficial versus detrimental roles of senescent cells throughout the lifespan in various physiological contexts, such as fracture healing, is of considerable importance (*Khosla et al., 2020*).

Similar to other tissues, senescent cells accumulate within the aged bone microenvironment and are causal in the pathogenesis of age-related bone loss (*Farr et al., 2016*; *Farr et al., 2017*; *Farr and Khosla, 2019*). One key feature of the cellular senescence program is activation of the cyclin-dependent kinase inhibitors (CDKIs), *Cdkn2a$^{Ink4a}$* (*p16$^{Ink4a}$*) and *Cdkn1a$^{Cip1}$* (*p21$^{Cip1}$*), paralleled by apoptosis resistance via upregulation of senescent cell anti-apoptotic pathways (*Tchkonia et al., 2013*). In the murine bone marrow, predominantly *Cdkn2a$^{Ink4a}$* and to a lesser extent *Cdkn1a$^{Cip1}$* have been linked to aging in diverse cellular populations such as B- and T-cells, myeloid cells, osteoprogenitors, osteoblasts, and osteocytes (*Farr et al., 2016*). Along with increased CDKI expression, senescent cells develop a senescence-associated secretory phenotype (SASP), consisting of pro-inflammatory cytokines, chemokines, and extracellular matrix degrading proteins that can drive tissue dysfunction via paracrine and systemic intercellular signaling, and spread cellular senescence through a so-called 'bystander effect' (*Lagnado et al., 2021*; *Nelson et al., 2012*; *Xu et al., 2015a*). These detrimental effects can be alleviated by clearing senescent cells either genetically or by administration of senolytics – drugs that selectively kill senescent cells. For example, intermittent delivery of the senolytic cocktail, Dasatinib (D, a tyrosine kinase inhibitor) plus Quercetin (Q, a natural flavonoid) has been shown to eliminate senescent cells in old mice and in humans and, in preclinical studies, slows the onset of aging by preventing multiple co-morbidities to thereby extend healthspan (*Farr et al., 2016*; *Farr et al., 2017*; *Xu et al., 2015b*). These findings are now being translated into humans, and D + Q is currently undergoing human trials for safety and efficacy to prevent age-related skeletal deterioration (NCT04313634).

In youth, beneficial physiological functions of senescent cells and their SASP have been suggested in skin wound healing, as senescent fibroblasts and endothelial cells are recruited to sites of injury where they release SASP factors that, in turn, attract various cell populations (e.g. immune cells) to accelerate skin wound healing. Indeed, after an early appearance in the healing skin wound, senescent cells induce myofibroblast differentiation and secrete PDGF-AA, a crucial factor necessary for proper wound healing (*Demaria et al., 2014*). However, the dynamics of skin wound healing do not directly translate to the various bone healing repair phases (i.e. inflammatory, soft callus, hard callus, and remodeling phase). Indeed, bone has the exclusive ability to form scar-free tissue de novo, although this process is slower than the typical wound healing process (*Marsell and Einhorn, 2011*). Importantly, manipulation at each stage can change the course of natural fracture healing events, which has been demonstrated by our group previously (*Saul et al., 2019*; *Undale et al., 2011*). Whether senescent cells appear in the healing skeleton as they do in skin wound healing or how these cells modulate fracture repair is not known. Indeed, to the extent that senolytic drugs progress to clinical use, particularly in patients with osteoporosis, it is critical to understand whether senescent cells play a similar, beneficial role in bone fracture healing as appears to be the case in the skin (*Demaria et al., 2014*). For example, if clearance of senescent cells impairs the timecourse or final outcome of the fracture healing process, that would clearly adversely impact the potential use of senolytic therapies in patients with osteoporosis.

In this context, the aims of this study were to characterize the potential appearance of senescent cells during fracture healing and establish whether targeting cellular senescence with senolytics impacts facture healing dynamics. Herein, we specifically focus on fracture healing in young adult mice to avoid the confounding effects of senescence with aging. We first defined the transcriptional profiles of the healing callus and performed an in-depth characterization of cellular senescence at various timepoints. We then leveraged a *Cdkn2a$^{Ink4a}$*-luciferase reporter mouse model (*Cdkn2a$^{LUC}$*) (*Burd et al., 2013*) to further validate the appearance and disappearance of these cells in vivo. Finally, we treated young adult fractured mice with senolytics to evaluate whether clearance of senescent cells adversely affected fracture healing, which is a critical issue in the further development of senolytic therapies for osteoporosis.

## Results

### Cdkn1a$^{Cip1}$ and Cdkn2a$^{Ink4a}$ are induced during murine fracture healing

To evaluate transcriptome-wide changes underlying the different stages of fracture healing, we analyzed publicly available mRNA-seq data of murine femoral fracture sites as well as intact control femora at five specific time points (*Coates et al., 2019*). At each time point, post-fracture femora

displayed a distinct gene expression profile upon clustering (*Figure 1A*). Within this dataset, samples collected on day 7 and 14 diverged most from the freshly fractured and intact bone. To identify cellular processes affected by fracture- and healing-associated mRNA expression changes, we performed gene ontology analyses comparing intact controls to femora 14 days post-fracture. As expected, cellular processes related to cell proliferation as well as apoptosis were significantly enriched (*Figure 1B and C*). To detect key common regulators among those biological processes, components of the respective gene sets were overlapped. Interestingly, the cell cycle regulators *Cdkn1a$^{Cip1}$* and *Cdkn2a$^{Ink4a}$* were present in all gene lists (*Figure 1D*). Notably, the importance of Cdkn2a$^{Ink4a}$ and *Cdkn1a$^{Cip1}$* during aging-associated bone deterioration has been previously observed by our group (*Farr et al., 2016*; *Farr et al., 2017*). Next, gene expression kinetics of *Cdkn1a$^{Cip1}$*, *Cdkn2a$^{Ink4a}$*, as well as the proliferation marker *Mki67* were examined after the induction of femoral fracture (*Figure 1E*). *Cdkn1a$^{Cip1}$* increased significantly very early following fracture (0.2 days), with a subsequent reduction, although at all time points its expression remained higher than levels in the intact contralateral femur. By contrast, *Cdkn2a$^{Ink4a}$* increased progressively up to day seven post-fracture and then plateaued. *Mki67* expression gradually decreased during bone healing (*Figure 1E*), indicating a decrease in proliferation. Together, these findings suggest a potential function of *Cdkn2a$^{Ink4a}$* and *Cdkn1a$^{Cip1}$* in regulating key cellular processes during fracture healing in mice with an emphasis on the late callus building phase around day 14.

## Senescent cells appear at fracture sites

For validation and extension of these in silico findings, we next induced femoral fractures in mice to further elucidate the possible roles of *Cdkn1a$^{Cip1}$* and *Cdkn2a$^{Ink4a}$* in fracture healing. After our in silico approach pointed to the end of the second week as the potential time point of highest senescent cell burden, we determined *Cdkn1a$^{Cip1}$* and *Cdkn2a$^{Ink4a}$* expression levels in the callus as well as the (intact) contralateral side after 4, 8, 14, and 28 days (*Figure 2A*). Notably, we detected a 20-fold increase in *Cdkn1a$^{Cip1}$* and a 100-fold increase in *Cdkn2a$^{Ink4a}$* 14 days post fracture compared to the contralateral side (*Figure 2B*). Due to the major function of *Cdkn1a$^{Cip1}$* and *Cdkn2a$^{Ink4a}$* in regulating cellular senescence (*Farr et al., 2016*; *Tchkonia et al., 2013*), we next tested for the presence of the most specific markers for cellular senescence by assessing senescence-associated distension of satellites (SADS) and telomere-associated DNA-damage foci (TAF) within the callus (*Farr and Almeida, 2018*; *López-Otín et al., 2013*). For this purpose, we performed immuno-fluorescent in situ hybridization (FISH) in the callus area after fracturing and stabilizing femora with an intramedullary pin. This analysis demonstrated that the number of SADS-positive senescent cells (*Farr et al., 2016*) in the callus was higher by ~6 -fold with a peak on days 8 and 14 relative to the contralateral control (*Figure 2C and D*). In agreement with this finding, the fractured side contained more TAF-positive senescent cells (*Figure 2E and F*). Indeed, the number of TAF-positive senescent cells was highest in the fracture callus on day 14 and higher than in the contralateral side throughout the entire observation period (*Figure 2F*), mirroring the changes in senescence genes and SADS over the time course. In summary, the murine callus displays a profound induction of *Cdkn1a$^{Cip1}$* and *Cdkn2a$^{Ink4a}$* expression as well as specific cellular senescence markers (SADS and TAF) 14 days after a femoral fracture.

## Cdkn2a$^{Ink4a}$-expressing cells accumulate in the callus in vivo and homozygous deletion of Cdkn2a$^{Ink4a}$ increases callus volume

After observing the induction of senescence and upregulation of *Cdkn2a$^{Ink4a}$* in the fracture callus, we sought to confirm the accumulation of *Cdkn2a$^{Ink4a}$* -expressing cells in vivo at the fracture site and evaluate the effects of homozygous deletion of *Cdkn2a$^{Ink4a}$* on fracture healing. For this purpose, we utilized *Cdkn2a$^{LUC}$* mice, in which the firefly luciferase cDNA is knocked in into the translational start site of the endogenous *Cdkn2a$^{INK4a}$* locus (*Burd et al., 2013*). Thus, these mice serve as both reporter mice for *Cdkn2a$^{Ink4a}$* expression and, in the homozygous state, are a complete *Cdkn2a$^{Ink4a}$* knock out. Upon fracture induction, *Cdkn2a$^{Ink4a}$*-positive cells were visualized using bioluminescence (*Figure 3A and B*). *Cdkn2a$^{Ink4a}$*-driven luminescence peaked approximately 2 weeks after fracture induction and declined thereafter (*Figure 3C*). Notably, *Cdkn2a$^{Ink4a}$* luminescence kinetics mirrored the time course of *Cdkn2a$^{Ink4a}$* mRNA expression we observed in silico and ex vivo and also paralleled the changes in SADS and TAF we observed as described above (*Figures 1E and 2C–F*). We also analyzed the

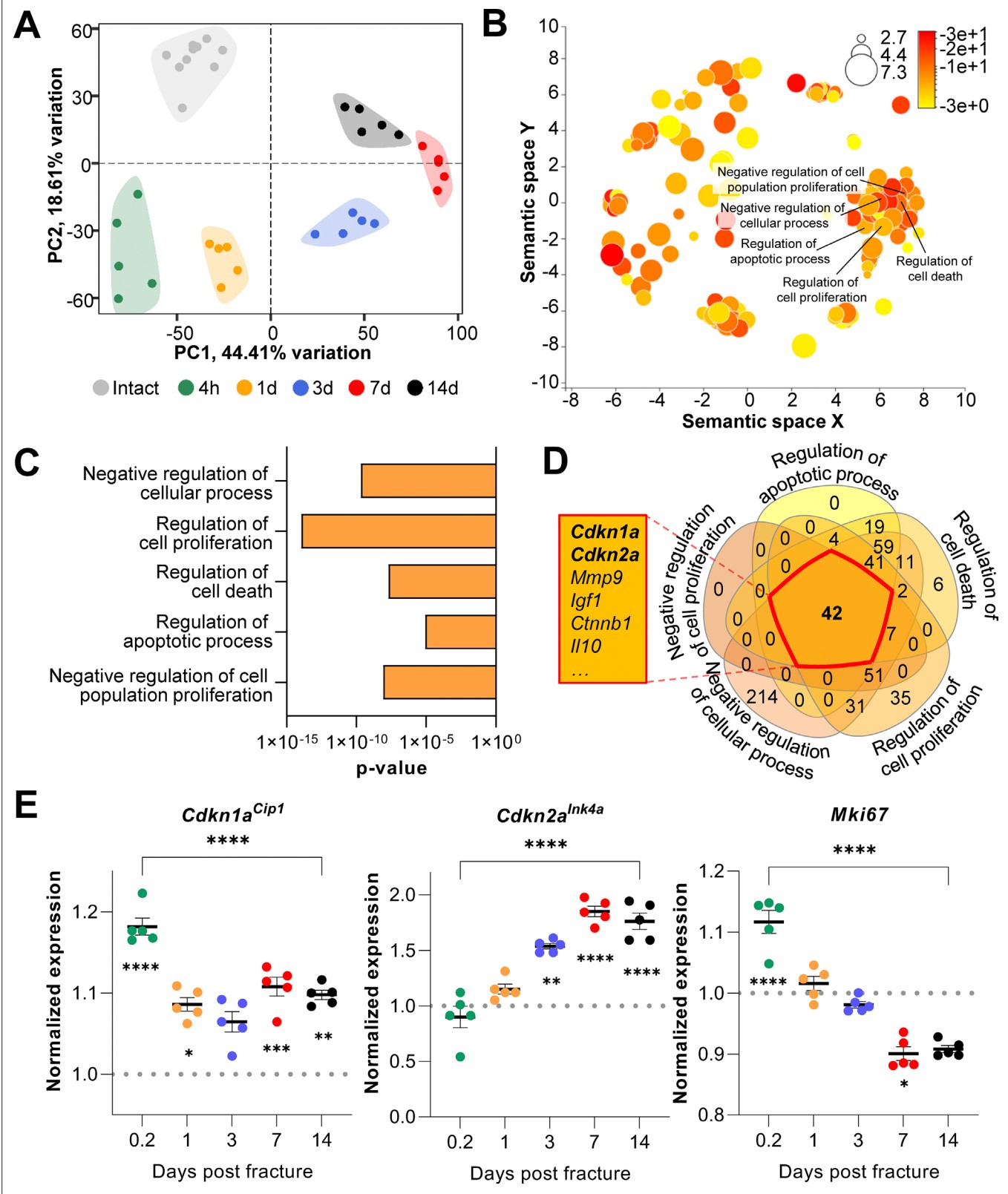

**Figure 1.** *Cdkn1a*[Cip1] and *Cdkn2a*[Ink4a] are upregulated after the induction of femoral fractures in mice. Transcriptome-wide changes during femoral fracture healing in mice were analyzed using publicly available mRNA-seq data (GSE152677 ***Coates et al., 2019***). (**A**) Principal component analysis (PCA) illustrates the variance of gene expression profiles when comparing intact controls to fractured femora (0.2, 1, 3, 7, and 14 days post fracture; n = 5 per time point). (**B**) Cellular pathways affected by gene expression changes 14 days after fracture compared to intact controls were detected using gene

*Figure 1 continued on next page*

*Figure 1 continued*

ontology (REVIGO; *Supek et al., 2011*). (**C**) REVIGO revealed that pathways associated with cell proliferation and cell death (GO:0048523, GO:0042127, GO:0010941, GO:0042981, GO: 0008285) were significantly affected 14 days after a mid-diaphyseal stabilized femoral fracture. (**D**) When overlapping these pathways, 42 genes were found to be present in all gene sets, including *Cdkn1a*<sup>Cip1</sup> and *Cdkn2a*<sup>Ink4a</sup>. (**E**) Normalized mRNA expression kinetics reveal an upregulation of *Cdkn1a*<sup>Cip1</sup> and *Cdkn2a*<sup>Ink4a</sup> during fracture healing while *Mki67* levels decreased. The dotted line represents the normalized baseline expression of intact control femora. The small asterisks indicate differences from the control. A-E: n = 35, all female. Mean ± SEM. One-way ANOVA; *p < 0.05, **p < 0.01, ***p < 0.001, ****p < 0.0001.

The online version of this article includes the following figure supplement(s) for figure 1:

**Source data 1.** Source data (normalized counts, REVIGO results, DE genes) for *Figure 1*, panels A-E.

---

consequences of *Cdkn2a*<sup>Ink4a</sup>-knockout on the time course of bone healing by weekly X-ray analyses performed according to *Wehrle et al., 2019*; *Figure 3D* and measurements of the relative callus area (*Figure 3E and F*). Although the overall time course of healing was not significantly altered in the *Cdkn2a*<sup>Ink4a</sup>-deficient mice, post-mortem end point analysis revealed that the callus bone volume was significantly increased in these mice relative to the wild-type mice (*Figure 3G*). These findings thus demonstrate that *Cdkn2a*<sup>Ink4a</sup>-positive cells accumulate at the fracture site in vivo and that complete loss of *Cdkn2a*<sup>Ink4a</sup> expression does not impair, but rather increases callus volume during fracture healing.

## Senescent cells and the SASP show distinct expression patterns in the fracture callus

Along with the appearance of cellular senescence, a locally detrimental bystander effect is caused by the *Cdkn1a*<sup>Cip1</sup>- and *Cdkn2a*<sup>Ink4a</sup>-positive cells via SASP secretion (*Jurk et al., 2014*; *Schafer et al., 2018*). As such, we performed a detailed analysis using qRT-PCR of the fracture callus over time. The SASP consists of, among others, growth factors and growth regulators (*Figure 4A*), out of which *Igfbp3* showed a 20-fold increase by day 14. Remarkably, the chemokine *Ccl7* was more than 70-fold increased on day 14 (*Figure 4B*). The protease Plasminogen activator inhibitor-1 (*Pai1* or *Serpine1*) was elevated by more than 60-fold on day eight but declined substantially on day 14 (*Figure 4C*). Several transcription factors, including *Nfkb1*, showed a modest increase, while *Foxo4* was significantly increased on days 8 and 14 (*Figure 4D*). In the TGFβ complex, *Tgfb1* itself displayed an initial increase and then a substantial decrease on day 28 which was even significantly lower than levels detected in the intact bone (*Figure 4E*). Several members of the interleukin family such as *Il6* and *Il17a* showed the typical rise within the first days of the fracture (inflammatory phase), whereas *Il1b* was downregulated (*Figure 4F*). Overall, the expression pattern of all senescence-associated and SASP genes was highly heterogeneous (*Figure 4—figure supplement 1*). However, distinct gene clusters displaying similar patterns could be constructed. For example, *Cdkn2a*<sup>Ink4a</sup> and *Cdkn1a*<sup>Cip1</sup> were detected within a cluster consisting of growth factors (*Igfbp3*, *Igfbp4*, *Vegfa*), proteases (*Mmp13*), and transcription factors (*Foxo4*). The similarity in regulation was indicated by their high correlation coefficients (*Figure 4G*). To determine which cell population may be associated with SASP secretion, we analyzed a publicly available single cell dataset (*Baryawno et al., 2019*, *Figure 4—figure supplement 2A, B*). After evaluating key regulators within murine bone and marrow, we found the highest enrichment of SASP genes within the mesenchymal stem cell (MSC) cluster (*Figure 4—figure supplement 2B-F*), which helped guide the subsequent studies below.

## Senescence in MSCs can be partially rescued by senolytic treatment in vitro

Based on the above experimental and in silico analyses, we next aimed to determine the effects of senolytics on MSCs in vitro. For this purpose, MSCs were isolated from murine bone marrow, cultured in vitro until confluent, and then treated with D + Q. The senolytic cocktail did not exert effects on cell numbers (*Figure 5A and B*). Subsequently, the cells were treated with 100 µM $H_2O_2$ to induce senescence (*Figure 5C and D*). Senescent cells were detected using SA-β-Gal staining and, as anticipated, addition of $H_2O_2$ resulted in profound cellular senescence (*Figure 5D*). Treatment with D + Q rescued this effect by significantly reducing the percentage of senescent cells from 33.1% to 13.1% (-60.3%; *Figure 5C and D*). Next, we treated mice with vehicle or D + Q in vivo and extracted MSCs to grow

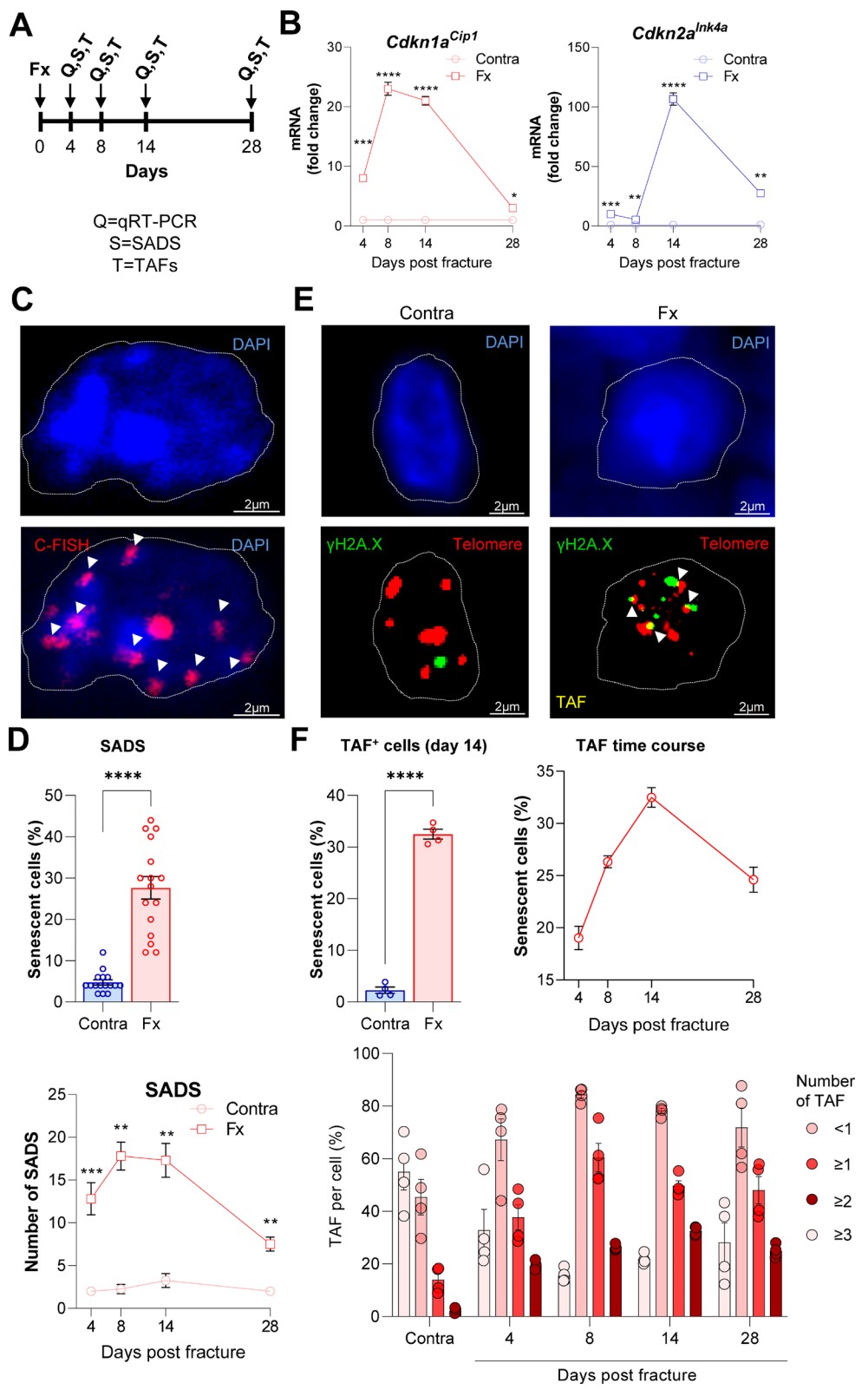

**Figure 2.** Cellular senescence is induced during femoral fracture healing. (**A**) Femoral fractures (Fx) were induced in male C57BL/6 mice with contralateral (Contra), intact control femora. Callus and controls were analyzed 4, 8, 14, and 28 days after fracture (n = 4 per time point). (**B**) *Cdkn1a^Cip1* and *Cdkn2a^Ink4a* mRNA expression levels during fracture healing were determined at each time point using rt-qPCR. *Cdkn1a^Cip1* was strongly induced 8–14 days

*Figure 2 continued*

and *Cdkn2a^Ink4a* 14 days post fracture. (**C**) Senescence-associated distension of satellites (SADS) were detected using immune-fluorescent in situ hybridization (FISH) Top: representative image of the contralateral side. Below: fractured side with four SADS. (**D**) The total number of SADS as well as percentage of SADS-positive cells ( ≥ 4 SADS/cell) at all time points was increased in the callus compared to the contralateral control femur. (**E**) Telomere-associated foci (TAF) were determined using FISH to detect telomeres (red) and immunofluorescent staining for γH2A.X (green). TAF (yellow, see arrowheads) were defined as sites of γH2A.X-associated DNA damage co-localized with telomeres (n = 70 cells were analyzed per bone). (**F**) The highest percentage of TAF were detected on day 14 of fracture healing. The number of TAF-positive senescent cells ( ≥ 3 TAF/cell) was increased in fractured bones throughout the entire healing period. B: n = 24 (n = 24 in Contra, n = 24 in Fx, n = 6 per timepoint), all male. C-D: n = 16 (n = 16 in Contra, n = 16 in Fx), all male. E-F: n = 20 (n = 4 per time point and n = 4 in contra on day 14), all male. Mean ± SEM. One-way ANOVA or Student's *t*-test for pairwise comparisons; *p < 0.05, **p < 0.01, ***p < 0.001, ****p < 0.0001.

The online version of this article includes the following figure supplement(s) for figure 2:

**Source data 1.** Source data (raw ct values, for *Figure 2*, panel B), raw numbers for SADS (panel D) and TAF (panel F).

them ex vivo (*Figure 5E*). The relative cell number of isolated MSCs was increased after 5 weeks of D + Q treatment in vivo (*Figure 5F*) while SA-β-Gal staining revealed a minor reduction in the number on senescent cells (–18.8%, *Figure 5G and H*). However, a priori in vivo treatment with D + Q did not significantly reduce the proportion of cells becoming senescent after $H_2O_2$ treatment (*Figure 5G and H*). Together, our in vitro studies suggest that $H_2O_2$-induced senescence in MSCs can be partially rescued by D + Q treatment.

## Pharmacological clearance of senescent cells does not impair but rather accelerates the time course of fracture healing in vivo

Based on our findings with the *Cdkn2a^Ink4a* knock out mice and D + Q effects on senescent MSCs in vitro, we hypothesized that, in contrast to skin wound healing (*Demaria et al., 2014*), clearance of senescent cells using D + Q treatment would enhance fracture healing in vivo. To test this, we treated wild-type mice with D + Q one day before fracture and on a weekly basis post fracture (*Figure 6A*). Based on our previous experiments where markers of senescence peaked on day 14 (*Figures 1E and 2B–F*), we sacrificed one mouse cohort at this time point and performed gene expression analysis of callus tissue. The senescence and SASP markers *Cdkn2a^Ink4a* and *Foxo4* displayed decreased mRNA levels in response to D + Q treatment, with a borderline decrease in *Pdgfa* (*Figure 6B*) along with additional SASP factors (*Figure 6—figure supplement 1A*); of interest, some factors (e.g., *Gdf15* and *Pdgfb*) increased following D + Q treatment (*Figure 6—figure supplement 1A*). The reduction of *Cdkn2a^Ink4A* mRNA was confirmed at the protein level (*Figure 6C*). To address the impact of D + Q treatment on fracture healing, weekly X-rays were performed and tibia healing was classified according to *Wehrle et al., 2019*. Notably, we detected significantly improved healing kinetics after the clearance of senescent cells with D + Q treatment (*Figure 6D*). Weekly two-plane assessments of the callus area confirmed an increased callus area 2 weeks after fracture (*Figure 6E and F*). In addition, the relative callus area was already reduced by week five in the D + Q group, indicating an accelerated callus restoration (i.e. earlier approach to the remodeling phase) after the maximum area had been reached (*Figure 6F*). Micro-CT analysis after 5 weeks demonstrated a higher callus volume in D + Q-treated mice (*Figure 6G*). Finally, we evaluated biomechanical properties of healed tibiae using torsional testing of the tibiae. In agreement with the elevated callus area and higher healing score on day 14, senolytic treatment resulted in a significant increase in the maximal torque that the healed bones were able to sustain before fracture, although the increase in bone stiffness did not achieve statistical significance (*Figure 6H*). Thus, D + Q treatment reduced gene expression levels of senescence markers/SASP markers and, in contrast to findings in skin wound healing (*Demaria et al., 2014*), did not impair but rather accelerated the time course of fracture healing.

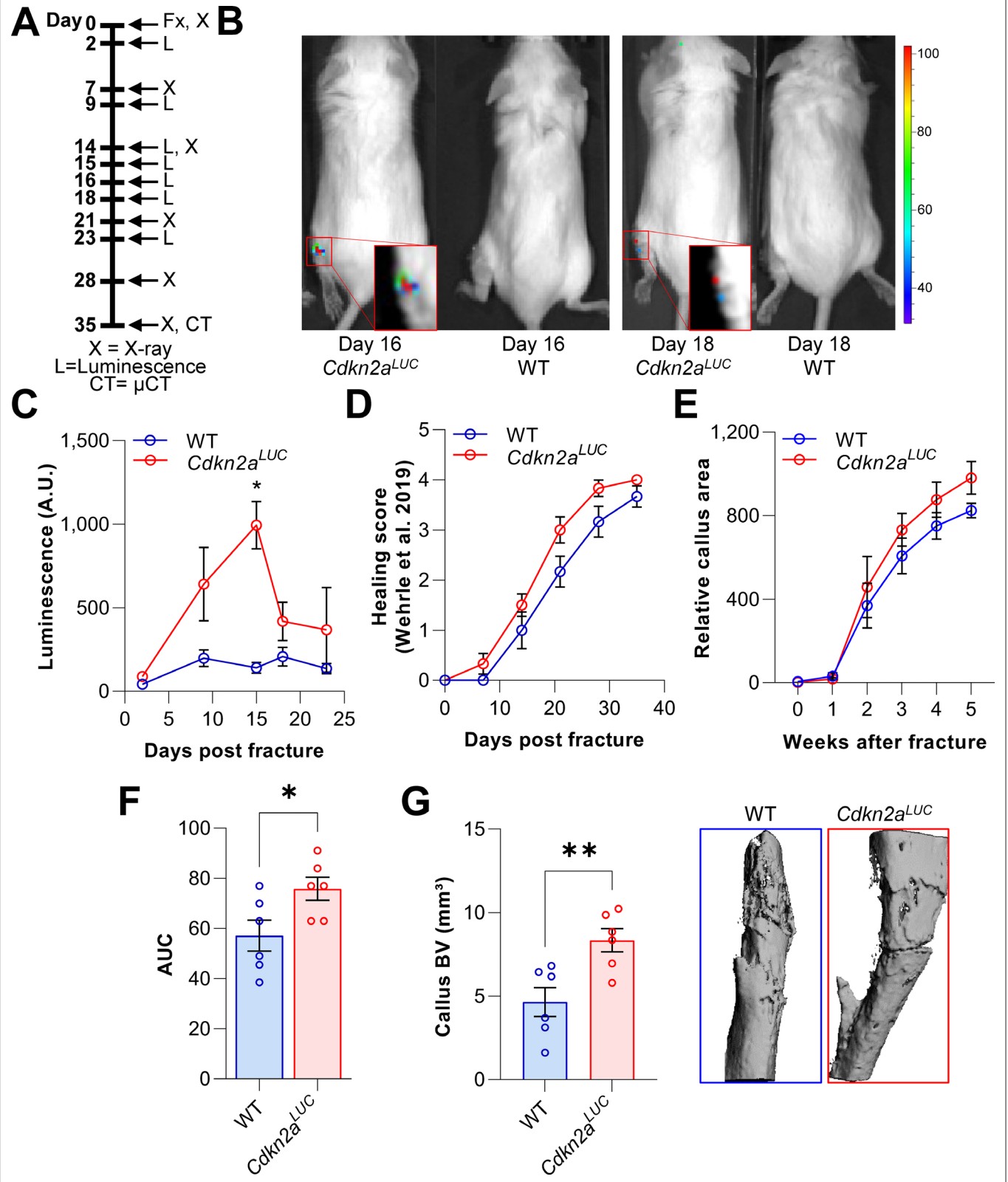

**Figure 3.** *Cdkn2a^{Ink4a}*-positive cells are present at the fracture site and negatively affect callus formation. The involvement of *Cdkn2a^{Ink4a}*-expressing cells in callus was evaluated in *Cdkn2a^{LUC}* or wild type (WT) mice. (**A**) Tibial fractures (Fx) were induced and *Cdkn2a^{Ink4a}*-dependent luminescence was analyzed between 2 and 23 days after fracture using Xenogen. X-rays were performed weekly and callus was isolated at day 28 for subsequent rt-qPCR. (**B**) Representative Xenogen images of *Cdkn2a^{Ink4a}*-associated luminescence at fracture sites in *Cdkn2a^{LUC}* mice. (**C**) Luminescence corresponding to

*Figure 3 continued on next page*

*Figure 3 continued*

*Cdkn2a^Ink4a*-expression after tibial fracture in WT and *Cdkn2a^LUC* mice as detected using Xenogen. *Cdkn2a^Ink4a*-associated luminescence reached its maximum 15 days after fracture while no signal was detected in WT controls. (**D**) Bone healing was scored based on X-ray analysis according to Wehrle and colleagues (***Wehrle et al., 2019***). Healing kinetics were similar among genotypes. (**E**) Relative callus area was quantified after X-ray analysis using FIJI. (**F**) *Cdkn2a^LUC* mice displayed an increase in callus area as determined by calculating the area under the curve (AUC, p = 0.036). (**G**) Callus bone volume (BV) was measured and visualized postmortem using micro-CT. The depletion of *Cdkn2a^Ink4a*-expressing cells resulted in significantly elevated callus BV. B-C: 6 WT (3 males, 3 females), 8 *Cdkn2a^Luc* (4 males, 4 females), D-F: 6 WT (3 males, 3 females), 6 *Cdkn2a^Luc* (3 males, 3 females), G: 6 WT (3 males, 3 females), 6 *Cdkn2a^Luc* (3 males, 3 females). Mean ± SEM. One-way ANOVA or Student's *t*-test for pairwise comparisons; *p < 0.05, **p < 0.01.

The online version of this article includes the following figure supplement(s) for figure 3:

**Source data 1.** Source data (raw AU values, for ***Figure 3***, panel B, **C**), raw scores (panel D), area (panel E), AUC (panel F), and BV (panel G).

## Discussion

Cellular senescence is generally considered in the context of its adverse effects during aging (***Baker et al., 2011***). Clearance of senescent cells has been shown to reverse age-associated symptoms and diseases in mice and senolytic therapies are now in early phase clinical trials for a range of age-associated disorders (***Khosla et al., 2020***; ***Palmer et al., 2021***; ***Tchkonia et al., 2021***). Similarly, clearance of *Cdkn2a^Ink4a*-positive senescent cells using genetic approaches has been shown to reduce age-associated diseases and extend healthspan in mice (***Braun et al., 2012***; ***Chandra et al., 2020***; ***Farr et al., 2019***; ***Saul and Kosinsky, 2021***; ***Xu et al., 2018***). However, previous studies on skin wound healing have suggested a beneficial role of transiently appearing senescent cells in that process (***Demaria et al., 2014***). This, in large part, has formed the basis for the hypothesis that senescent cells, although detrimental in the context of aging, may be beneficial for tissue repair via the secretion of the SASP, in part by attracting immune cells and initiating the tissue repair process (***Kowald et al., 2020***). However, whether this concept holds true across tissues beyond the skin is of considerable importance, particularly for the potential translation of senolytic therapies for osteoporosis. Indeed, a detrimental effect of senescent cell clearance on fracture repair would pose a substantial roadblock for the development of senolytic therapies for osteoporosis. Relevant to this issue, we demonstrate senescent cells in the dynamically healing bone and define the time course of appearance and subsequent disappearance of these cells from the fracture callus. Importantly, using both a genetic and pharmacological model, we reduce the senescent cell burden and demonstrate no adverse effects, but rather beneficial effects (i.e. increased callus volume in the *Cdkn2a^Ink4a* knock out model and accelerated timecourse of healing with senolytics) on fracture healing.

As noted earlier, we focused this initial study on young adult mice in order to avoid potential confounding due to aging and to define the possible physiological role of senescent cells in fracture healing. At a gene expression level and using two independent post-transcriptional techniques (SADS and TAF), which are the most specific markers for cellular senescence (***Anderson et al., 2019***; ***Farr et al., 2016***; ***Hewitt et al., 2012***) we were able to first describe the appearance of senescent cells within the physiologically healing murine bone. Moreover, we established a timecourse of cellular senescence peaking on day 14 in healing long bones and made use of a genetic knock out model of *Cdkn2a^Ink4a*, the *Cdkn2a^LUC* mouse (*19*), to both visually detect and assess the role of *Cdkn2a^Ink4a* knock out on fracture healing. In line with the findings by Demaria et al. in skin wound healing (***Demaria et al., 2014***), a transient increase of *Cdkn2a^Ink4a*-positive cells was detected in the final stages of the anabolic and beginning catabolic phase of fracture healing. The healing process itself, however, was not fundamentally changed by the functional reduction of *Cdkn2a^Ink4a* but the callus volume was significantly enhanced in mice with *Cdkn2a^Ink4a* deletion relative to wild-type controls.

Our analysis of publically available single-cell RNAseq data (***Baryawno et al., 2019***) indicated that MSCs in murine bone and bone marrow are an important cell population undergoing senescence and we further demonstrated effects of D + Q in clearing senescent MSCs in vitro. However, the precise identity of the cells undergoing transient senescence in the fracture callus remains unclear. This is particularly important in the fracture callus, as there are several stages of fracture healing (***Claes et al., 2012***; ***Marsell and Einhorn, 2011***), including formation of a hematoma (~days 1–5), fibrocartilagenous phase (~days 5–11), bone formation (~days 11–18), and bone remodeling (after day 18); thus, multiple other cell types (e.g. cartilage cells, others) in

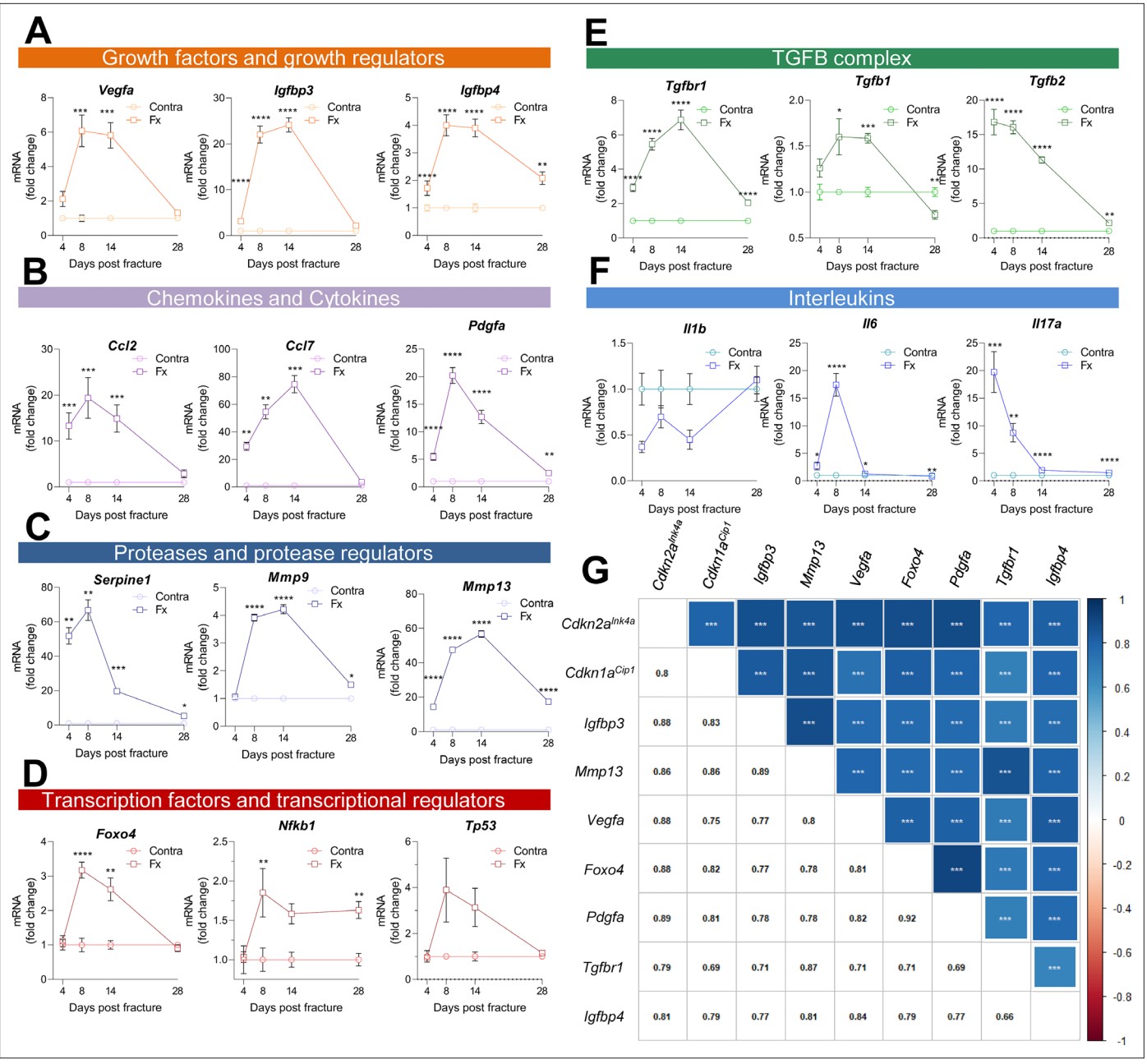

**Figure 4.** Multiple SASP factors show a marked increase in the healing callus, while distinct genes form similarly behaving clusters. SASP-associated factors can be subdivided into functional subunits. (**A**) mRNA expression of the growth factors *Vegfa*, *Igfbp3*, and *Igfbp4* substantially increased during the callus forming phase (days 2–15). (**B**) The chemokines *Ccl2* and *Pdgfa* more profoundly rose in gene expression levels in the soft callus (days 2–7) but not in the hard callus (days 8–14) phase, while *Ccl7* was more than 70-fold increased on day 14. (**C**) Expression of the protease *Serpine1* was elevated in the soft callus phase, similar to *Pdgfa*, while *Mmp9* and *Mmp13* peaked on day 14. (**D**) The transcription factors *Foxo4*, *Nfkb1*, and *Tp53* peaked on day 8, marking the beginning of hard callus formation. (**E**) TGFβ-associated genes displayed heterogeneous expression patterns. While *Tgfbr1* peaked on day 14, *Tgfb1* reached a plateau on days 8 and 14, and *Tgfb2* gradually declined after an initial peak on day 4. (**F**) Among the interleukins, *Il1b* was negatively regulated in the beginning of the healing phase, while *Il6* shortly peaked in the inflammatory and soft callus phase. *Il17a* showed a gradually decline, comparable to *Tgfb2*. (**G**) The largest cluster displaying similar gene expression patterns among all senescence-associated and SASP-gene markers included the key cell cycle regulators *Cdkn2a^Ink4a* and *Cdkn1a^Cip1*. The square size is proportional to the correlation coefficient, which is also depicted in the left bottom corner. Significant results were indicated with asterisks. A-G: n = 24 (n = 24 in Contra, n = 24 in Fx, 6 per time point per group), all male. Mean ± SEM. Multiple *t*-test (FDR); *$p < 0.05$, **$p < 0.01$, ***$p < 0.001$, ****$p < 0.0001$.

The online version of this article includes the following figure supplement(s) for figure 4:

**Source data 1.** Source data (raw ct values, for *Figure 4*, panel A-F), raw matrix (panel G).

*Figure 4 continued on next page*

*Figure 4 continued*

**Figure supplement 1.** Matrix of all SASP-associated genes and the two senescence key regulators Cdkn2a$^{Ink4a}$ and Cdkn1a$^{Cip1}$.

**Figure supplement 1—source data 1.** Raw matrix of the calculated matrix.

**Figure supplement 2.** Single-cell analysis of murine bone and bone marrow unveiling a potential origin of SASP secreting cells.

addition to MSCs could be undergoing senescence, and characterizing these senescent cells is the focus of ongoing work in our laboratory.

The role of the SASP in the skeletal system has mostly been shown to be adverse (*Farr and Khosla, 2019*; *Millerand et al., 2019*; *Yao et al., 2020*). We demonstrate here a substantial increase in the SASP during fracture healing, with some SASP factors increasing during the inflammatory phase and others in early and late callus forming phases. Although multiple approaches have been used to subclassify the SASP (*Acosta et al., 2013*; *Basisty et al., 2020*; *Coppé et al., 2010*; *Özcan et al., 2016*), we for the first time elucidated its components within healing bone and found specific *Cdkn2a$^{Ink4a}$* and *Cdkn1a$^{Cip1}$* clusters of associated SASP factors with a concordant timecourse during fracture healing. Elucidating the potential therapeutic applicability of these findings, we made use of a first-generation senolytic cocktail, D + Q. This treatment has successfully moved from preclinical studies after reducing the cellular senescence burden in bone (*Farr et al., 2017*) and adipose tissue (*Xu et al., 2018*) and is currently in multiple clinical trials, including for the prevention and/or treatment of osteoporosis (NCT04313634). We demonstrated its efficiency in reducing MSC senescence in vitro and treated healing murine long bones in vivo with this senolytic regimen. Following senolytic treatment, the senescence marker *Cdkn2a$^{Ink4a}$* was substantially reduced, along with a number of SASP markers. The time course of bone healing was accelerated, and biomechanical bone parameters partly enhanced (i.e. maximal torque). Notably, some SASP markers, specifically *Gdf15* and *Pdgfb*, were increased after the senolytic treatment. It has been hypothesized that up to a certain threshold, a transient senescent burden may be favoring healing (e.g. skin wounds), while above this threshold there may be a failure of wound closure, as observed in chronic wounds (*de Magalhães and Passos, 2018*; *Pratsinis et al., 2019*; *Wilkinson and Hardman, 2020*). Our findings suggest that cellular senescence in healing bone follows a transient time course that peaks around the second week of fracture healing, thereby potentially suppressing callus formation. Given that senolytics do not kill all senescent cells, but rather reduce their proportion (e.g. by ~30 % *Zhu et al., 2015*) our data demonstrate that this reduction is sufficient to enhance callus formation and at least does not impair the biomechanical properties of the healed bone.

In addition to implications for senolytic therapies for osteoporosis, our findings also have potential biological implications. Specifically, the concept of senescent cells as physiologically facilitating tissue repair which, to date, has been definitively demonstrated principally in the skin (*Demaria et al., 2014*) may be overly simplistic. Thus, contrary to the skin and uniquely in the living organism, bone has the ability to fully recover without losing its integrity and form a scar. As such, it is possible that the transient appearance of senescent cells is an evolutionarily conserved mechanism during injury repair, with beneficial effects on skin wound healing, which is associated with scar formation. By contrast, the lack of scar formation in bone and the already highly inflammatory state following fracture (*Marsell and Einhorn, 2011*) may render these transiently appearing senescent cells less useful and, as demonstrated in our study, potentially detrimental, to injury repair in bone. Clearly, further studies are needed to test this hypothesis, but our findings should stimulate a reconsideration of senescent cells as generally beneficial in the setting of tissue injury and repair.

In summary, we demonstrate that cellular senescence is present in fracture callus development using both transcriptional and more specific analyses evaluating distension of satellite heterochromatin and telomeric DNA damage, hallmarks of cellular senescence (*López-Otín et al., 2013*). We established the time course of *Cdkn2a$^{Ink4a}$* or *Cdkn1a$^{Cip1}$* expression during fracture healing, associated well-described SASP components to these key senescent genes, and reduced the senescent burden with senolytics. Importantly, this approach did not impair, but rather enhanced the fracture healing process in vivo. Collectively, our findings have clinical implications for the development of senolytic therapies for osteoporosis and also have biological relevance for our concept of senescent cells as

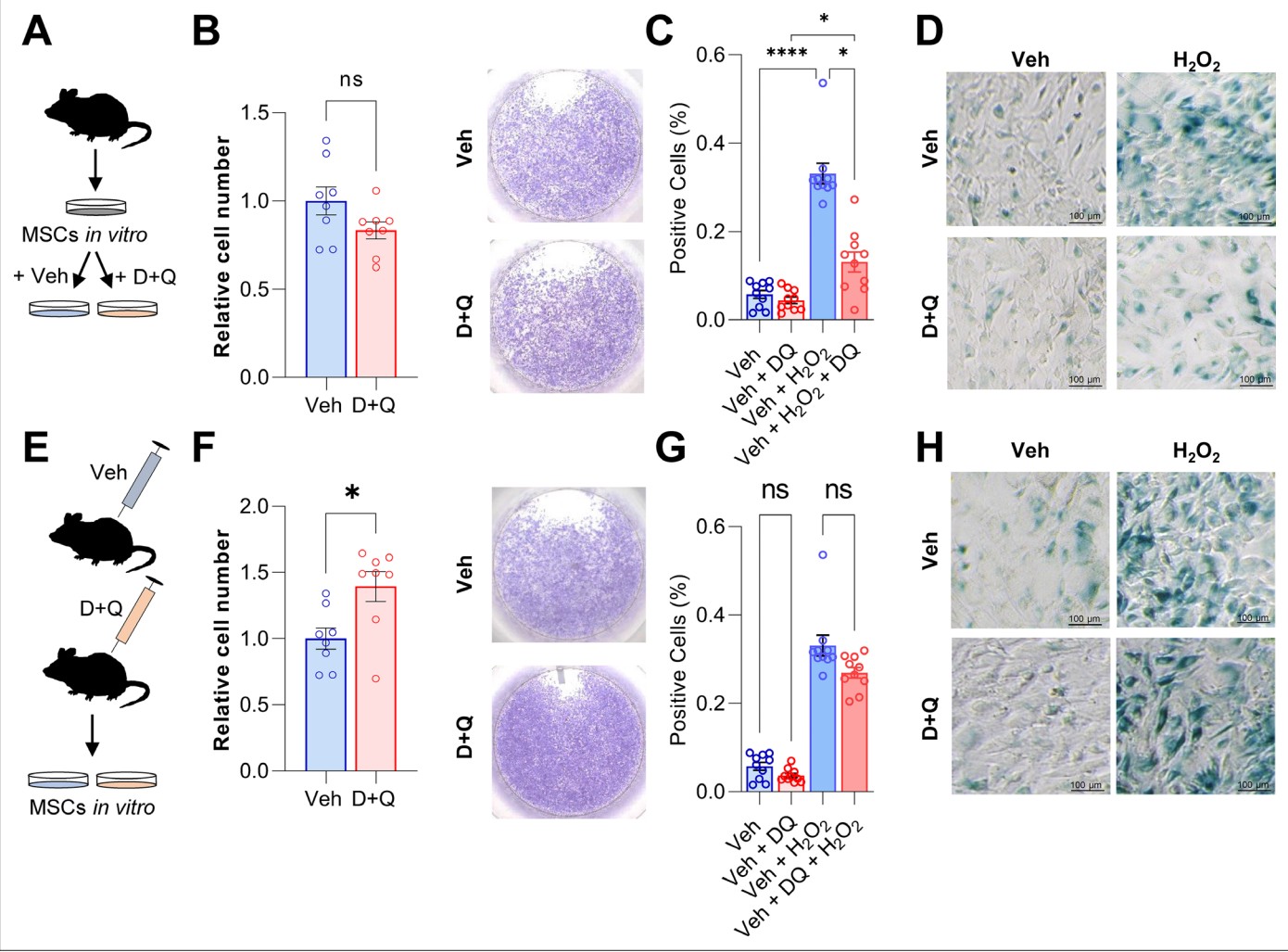

**Figure 5.** D + Q treatment can partially rescue senescence in MSCs in vitro. (**A**) Bone marrow MSCs were isolated from wild-type mice and grown in vitro until a confluence of approximately 70 % was observed. (**B**) After seeding the same cell numbers, MSCs were incubated with 200 nM Dasatinib and 50 μM Quercetin (D + Q) or vehicle solution for 24 hr. Relative cell numbers were evaluated 24 hr after D + Q or vehicle treatment (n = 6 per treatment). Besides using an automated cell counter, cells were stained using crystal violet to visualize cell densities. D + Q treatment did not change relative cell numbers. (**C**) MSCs were treated with 100 μM $H_2O_2$ for 4 hr to induce senescence, washed with PBS and cultured for another three days. Afterwards, cells were treated with D + Q or vehicle solution for 24 hr. After fixation, senescent cells were visualized using SA-β-Gal staining and quantified using FIJI (n = 4). (**D**) Representative images of SA-β-Gal staining. While $H_2O_2$-induced senescence, this effect was rescued by subsequent D + Q treatment. (**E**) Wild type mice were treated with 5 mg/kg BW Dasatinib and 50 mg/kg body weight Quercetin or vehicle via oral gavage weekly for 5 weeks. Bone marrow MSCs were isolated and cultured in vitro. (**F**) Relative cell number of D + Q-treated mice was increased 48 hr after seeding the same cell numbers (n = 6). Cell confluence was visualized using crystal violet staining. (**G**) In vivo treatment with D + Q resulted in a minor reduction in the number on senescent cells. $H_2O_2$ treatment induced cellular senescence in a comparable manner to the previous approach (**C**). (**H**) Representative images of SA-β-Gal staining. An a priori in vivo treatment with D + Q did not significantly reduce the proportion of cells becoming senescent after $H_2O_2$ treatment. A-H: n = 16, all male (n = 8 in Veh, n = 8 in D + Q). Mean ± SEM. One-way ANOVA or Student's *t*-test for pairwise comparisons; *p < 0.05, **p < 0.01, ***p < 0.001, ****p < 0.0001.

The online version of this article includes the following figure supplement(s) for figure 5:

**Source data 1.** Raw cell numbers for **Figure 5**, panel B, F, raw cell numbers (panel C, **G**).

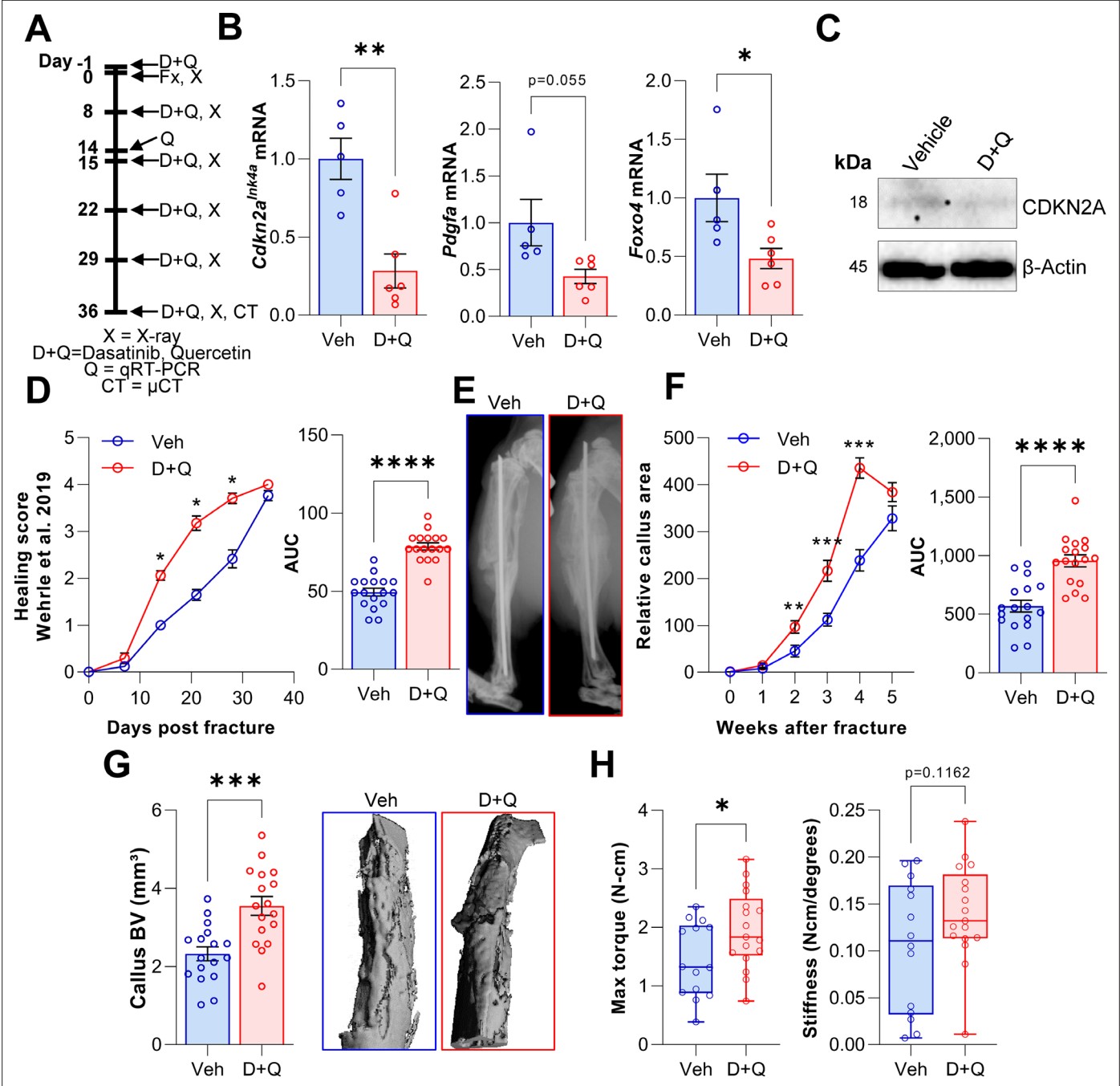

**Figure 6.** D + Q treatment accelerates the time course of fracture healing. (**A**) Senescent cells were cleared by oral gavage of wild type mice one day before tibial fracture and weekly with 5 mg/kg BW Dasatinib plus 50 mg/kg BW Quercetin or vehicle solution(B) mRNA levels of the senescence and SASP markers *Cdkn2a*$^{Ink4a}$, *Pdgfa* and *Foxo4* were reduced by D + Q treatment 14 days after fracture induction as detected using rt-qPCR. (**C**) Western blot analysis of callus material revealed reduction of Cdkn2a$^{INK4A}$ protein level in response to D + Q. (**D**) Bone healing was assessed using X-ray analysis according to ***Wehrle et al., 2019***. Bone healing was significantly improved with D + Q. (**E**) Representative X-ray images of vehicle- and D + Q-treated mice 14 days after tibial fracture. Callus volume appeared to be increased with D + Q. (**F**) Relative callus area was detected after X-ray analysis and quantification using FIJI, showing an acceleration in the time course of callus formation. (**G**) Callus bone volume at 5 weeks determined using μCT was increased in D + Q-treated mice. (**H**) Biomechanical analysis of the tibiae revealed a significant increase in maximum torque upon senolytic treatment and a non-significant increase in bone stiffness. Mean ± SEM. B, C: n = 11 (5 Veh [3 males, 2 females], 6 DQ [3 males, 3 females]), D-G: n = 34, all females (17 Veh, 17 DQ), H: n = 31, all females (14 Veh, 17 DQ; note that three bones from the Veh group could not be analyzed due to technical issues in preparing these bones for torsional testing). For B: Univariate linear model with the covariate 'sex', D-H: One-way ANOVA or Student's *t*-test for pairwise comparisons; *p < 0.05, **p < 0.01, ***p < 0.001, ****p < 0.0001.

*Figure 6 continued on next page*

*Figure 6 continued*

The online version of this article includes the following figure supplement(s) for figure 6:

**Source data 1.** Raw ct values for *Figure 6*, panel B.

**Source data 2.** Raw uncropped and annotated blot images for *Figure 6*, panel C.

**Figure supplement 1.** Changes of SASP factors in the fracture callus.

**Figure supplement 1—source data 1.** Raw ct values for *Figure 6—figure supplement 1*, panel A.

facilitating healing, as these beneficial versus detrimental effects of senescent cells on injury repair may vary across tissues.

# Materials and methods

## Key resources table

| Reagent type (species) or resource | Designation | Source or reference | Identifiers | Additional information |
|---|---|---|---|---|
| Strain, strain background (*Mus musculus*) | C57BL/6 | Charles River Laboratories | RRID:IMSR_CRL:027 | |
| Strain, strain background (*Mus musculus*) | Alb.B6.Cdkn2a-Luciferase | Sharpless Lab, North Carolina, USA | RRID:IMSR_NCIMR:01XBT | |
| Cell line (*Mus musculus*) | MSC (normal, adult) | This paper | - | freshly isolated from C57BL/6 mice |
| Antibody | γ-H2A.X, anti-rabbit antibody | Cell Signaling | RRID:AB_2118009 | "(1:200)" |
| Antibody | goat, anti-rabbit antibody | Vector Laboratories | RRID:AB_2313606 | "(1:200)" |
| Antibody | Cy5 Streptavidin, anti-rabbit antibody | Vector Laboratories | RRID:AB_2868518 | "(1:500)" |
| Other | TelC-Cy3 | PNA Bio | RRID:AB_2893285 | (PNA) probe |
| Other | CENPB-Cy3 | PNA Bio | RRID:AB_2893286 | (PNA) probe |
| Antibody (WB) | Cdkn2aINK4A | Abcam | RRID:AB_2891084 | ("1:1,000") |
| Antibody (WB) | Beta-Actin | Cell Signaling | RRID:AB_2223172 | ("1:1,000") |
| Chemical compound, drug | Xenolight RediJect Coelentarazine h | Calipers | #760,506 | |
| Software, algorithm | GraphPad Prism software | San Diego, CA | RRID:SCR_002798 | |
| Software, algorithm | R 4.0.3 | San Diego, CA | RRID:SCR_001905 | |
| Other | DAPI stain | Life Technologies | RRID:AB_2629482 | |

## Animal studies

Animal studies were performed under protocols approved by the Institutional Animal Care and Use Committee (IACUC), and experiments were performed in accordance with Mayo Clinic IACUC guidelines. All assessments were performed in a blinded fashion. Mice were housed in ventilated cages and maintained within a pathogen-free, accredited facility under a 12-hr light/dark cycle with constant temperature (23 °C) and access to food (diet details are specified below) and water ad libitum. We used young adult C57BL/6 N mice (starting at four months, ending at 5–6 months of age) for our experimental procedures. For anesthesia during surgery, we used isoflurane (vaporizer, 1.5–2% in oxygen, inhalation) for induction and maintenance until the surgery was complete (about 20 min).

A schematic of the study design in each of the three study phases (WT, Cdkn2a$^{LUC}$, D + Q) and respective timelines for the treatments is shown in *Figures 2A, 3A and 6A*.

For the first study (WT), four-month-old male C57BL/6 N WT mice (n = 24, all male) of comparable mean body weights received a standardized, closed diaphyseal femoral fracture. After a lateral parapatellar knee incision, the left femur was exposed while the tendons and muscles were protected.

After that, a transverse osteotomy with a rotary bone saw was introduced. An Insect Pin (Fine Science Tools, 26001–30, Austerlitz Insect Pin rod diameter 0.03 mm) was inserted retrogradely from the trochlear groove to stabilize the transverse femoral shaft fracture. After wound closure, postoperative pain management was performed with subcutaneous buprenorphine hydrochloride (0.1 mg/kg body weight [BW]) and the correct position of the pin was immediately affirmed by X-ray. Normal weight bearing postoperatively was allowed. X-rays were performed on a weekly basis as described in Radiographical fracture healing assessment. For each timepoint (4, 8, 14, 28 days), n = 6 male mice per time point were used (n = 24 mice, all male).

For the *Cdkn2a^{LUC}* study, Alb.B6.Cdkn2a-Luciferase mice were used as described in detail elsewhere (**Burd et al., 2013**). *Cdkn2a^{LUC}* mice were used: Alb.B6.Cdkn2a-Luciferase mice (n = 6 *Cdkn2a^{WT/WT}* (3 males, 3 females) and n = 8 *Cdkn2a^{LUC/LUC}*(4 males, 4 females), herein referred to as WT and *Cdkn2a^{LUC}*, respectively) with a homogenous Alb.B6.Cdkn2a^{Luc/Luc} and their littermates were used. Here, we performed a left transverse tibia fracture. The mid-shaft tibia was incised along the anterior side of the tibia, and an unilateral transverse osteotomy with a rotary bone saw was performed. The fractured bone was stabilized with an intramedullary Insect Pin was retrogradely implemented (**Wang et al., 2016**). X-rays and pain management was performed as described in the WT study. In addition to weekly X-rays, luminescence was detected (see Bioluminescence). The bioluminescence assay was performed in n = 8 mice, while two additional mice received weekly X-rays.

For the D + Q study, 4-month-old female C57BL/6 mice (n = 34 in total) were randomized to either vehicle (Veh, n = 17) or Dasatinib plus Quercetin (D + Q, n = 17) treatment on a weekly basis (**Figure 6A**). Vehicle or D + Q was administered via oral gavage. The weekly treatment regimen was verified to be effective in eliminating senescent cells by our group previously (**Farr et al., 2017**). Senolytics were diluted in 10 % EtOH, 30 % PEG, 60 % Phosal-50and administered via oral gavage at dosages of 5 mg/kg BW Dasatinib and 50 mg/kg BW Quercetin, respectively, in a total volume of 100 µl. The surgical procedure was performed as described above and weekly X-rays were carried out to indicate the fracture healing course. Postmortem, both intact and fractured tibiae were stored in 0.9 % saline (NaCl)-soaked gauze at –20 °C for direct scanning with ex vivo micro-computed tomography (micro-CT) (see Skeletal imaging) and subsequent biomechanical strength testing by standardized rotational testing (see torsional testing).

## Mouse tissue collection and assessments

Prior to sacrifice, body mass (g) was recorded and serum was collected in the morning under fasting conditions from anesthetized mice via cardiac puncture and stored at –80 °C. In the *Cdkn2a^{LUC}* and D + Q study, tibiae, femora, and humeri were excised from the mice and skeletal muscle/connective tissues were removed after euthanasia. The left tibia was stored in 0.9 % saline (NaCl) soaked gauze at –20 °C for direct ex vivo micro-computed tomography (micro-CT) scanning (see Skeletal imaging) and subsequent biomechanical strength testing by standardized torsional testing (see torsional testing). In the WT study, both non-decalcified femora were embedded in methyl-methacrylate and sectioned for immunohistochemistry (IHC) (see Skeletal histomorphometry assessments), and fluorescent in situ hybridization (FISH) (see Senescence-associated distension of satellites [SADS]).

In the D + Q study, the right tibia, both femora and both humeri were used to extract and culture bone marrow stem cells (BMSC; see Cell culture experiments). In the D + Q study and WT study, for fracture RNA analysis, the pin was removed, and the visually verified callus area removed after cleaning the bone from surrounding tissue. This callus area was minced with a scalpel in FACS buffer, and homogenized in QIAzol Lysis Reagent (QIAGEN, Valencia, CA), and frozen at –80 °C for real-time quantitative polymerase chain reaction (qRT-PCR) mRNA gene expression analyses (see Real-time quantitative polymerase chain reaction analysis).

## Skeletal imaging: radiographical fracture healing assessment

All imaging and analysis was performed in a blinded fashion as described by our group previously (**Undale et al., 2011**). Briefly, radiographs of the fractured femora or tibiae were taken under anesthesia after surgery and on a weekly basis. Therein, mice were in a supine position and both limbs extended. Anteroposterior (ap) and lateral (lat) planes were assessed. Radiographs were evaluated by two blinded researchers and scored for fracture healing using the score by **Wehrle et al., 2019** and **Undale et al., 2011**. After a satisfactory accordance was obtained (R² = 0.89, **Figure 6—figure**

*supplement 1B*), we decided to report the initially CT-based score by Wehrle et al. only, since the two planes gave us a good representation of all four cortices. The radiographs of fracture callus were quantified and analyzed with FIJI (NIH, Bethesda, MD, USA), as described elsewhere (*Liu et al., 2018*).

## Skeletal imaging: ex vivo micro-CT imaging

At study endpoint, callus volume of the left (fractured) tibia (proximal metaphysis to distal metaphysis) was performed. Scan settings were: 55 kVp, 10.5 μm voxel size, 21.5 diameter, 145mA, 300 ms integration time. For the callus volume measurement, a threshold of 190 and 450 were chosen according to the manufacturer's protocols (Scanco Medical AG, Basserdorf, Switzerland).

## Bioluminescence

In vivo luminescence was measured with a Xenogen IVIS Spectrum instrument and Living Image software (5 min medium binning). Bioluminescent signal quantification (photons/s/cm²/sr) of the fracture region was measured with the Living Image software. Mice were i.p. injected with 15 μg of Xenolight RediJect Coelentarazine h (Calipers). After 25 min, mice were anesthetized with isoflurane and imaging was performed.

## Torsional testing of tibiae

Tests of torsional load were performed in a blinded fashion. After removal of the pins and embedding of the tibial plateau as the distal tibia, the torsional load was applied at speed of 5°/s for a maximum of 36 s. Maximum rotation angle at failure (Deg) and stiffness (N-cm/degree) were used as primary endpoints. Maximum torque was the highest force that the bone could sustain before fracture, and stiffness was calculated from the linear portion of the loading curve (higher values for both are indicative of stronger bone) (*Undale et al., 2011*).

mRNA-sequencing (mRNA-seq) and analysis mRNA sequencing data was obtained from GSE152677 (*Coates et al., 2019*). Thirty-five female femoral fracture mRNA-seq datasets were aligned to the mouse reference genome mm10. The fastq-dump files were obtained from the BioProject PRJNA640062. Quality control of fastq files was carried out via FastQC (version 0.72) and reads were mapped to the mouse reference genome [mm10] using HISAT2 (version 2.1.0) on Galaxy (version 7). Read counts were generated using the featureCounts tool (version 2.0.1) and analyzed for differential gene expression using DESeq2 (version 2.11.40.6). Significantly differentially regulated genes were selected by a Benjamini-Hochberg adjusted *P*-value < 0.05 and log2-fold changes above 0.5 or below –0.5. The used R-packages for the downstream analysis were PCAtools (2.2.0) and corrplot (0.84).

## Analysis of published single-cell RNA-sequencing (scRNA-seq) data

Publicly available single cell RNA-seq data based on bone and bone marrow from C57BL/6 J mice (n = 4 each, all male) (*Baryawno et al., 2019*; *van Gastel et al., 2020*) (GSE128423) were analyzed. Sequencing data were aligned to the mouse reference genome mm10 and cells with at least 500 unique molecular identifiers (UMIs), log10 genes per UMI >0.8, > 250 genes per cell and a mitochondrial ratio of less than 0.2 % were extracted, normalized and integrated using the Seurat package v3.0 in R 4.0.2. Subsequent R-packages were Nebulosa (3.13) and Monocle (2.18.0).

## Quantitative real-time polymerase chain reaction (qRT-PCR) analysis

For callus analyses, callus and contralateral intact bone were removed as described in *Coates et al., 2019* and immediately homogenized in QIAzol Lysis Reagent (QIAGEN, Valencia, CA), and stored at –80 °C. Briefly, after soft tissue removal, a 7 mm section around the fracture site (and a 7 mm section at the same location on the intact contralateral site) was extracted and homogenized in QIAzol. Subsequent RNA extraction, cDNA synthesis, and targeted gene expression measurements of mRNA levels by qRT-PCR were performed as described previously (*Eckhardt et al., 2020*). Total RNA was extracted according to the manufacturer's instructions using QIAzol Lysis Reagent. Purification with RNeasy Mini Columns (QIAGEN, Valencia, CA) was subsequently performed. On-column RNase-free DNase solution (QIAGEN, Valencia, CA), was applied to degrade contaminating genomic DNA. RNA quantity was assessed with Nanodrop spectrophotometry (Thermo Fisher Scientific, Wilmington, DE). Standard reverse transcriptase was performed using High-Capacity cDNA Reverse Transcription Kit (Applied Biosystems by Life Technologies, Foster City, CA). Transcript mRNA levels were determined

by qRT-PCR on the ABI Prism 7900HT Real Time System (Applied Biosystems, Carlsbad, CA), using SYBR green (Qiagen, Valencia, CA). The mouse primer sequences, designed using Primer Express Software Version 3.0 (Applied Biosystems), for the genes measured by SYBR green are provided in *Supplementary file 1*. Input RNA was normalized using two reference genes (*Actb*, *Gapdh*) from which the most stable reference gene was determined by the geNorm algorithm. For each sample, the median cycle threshold (Ct) of each gene (run in triplicate) was normalized to the geometric mean of the median Ct of the most stable reference gene. The delta Ct for each gene was used to calculate the relative mRNA expression changes for each sample. Genes with Ct values > 35 were considered not expressed (NE), as done previously (*Farr et al., 2016*).

## Senescence-associated distension of satellites (SADS) analysis

Our group recently demonstrated that with fluorescence in situ hybridization (FISH), senescent cells display large-scale unraveling of peri-centromeric satellite heterochromatin DNA, referred to as SADS, a feature of senescent cells which has been demonstrated in osteocytes, fibroblasts, hepatocytes, glial cells and multiple other cell types (*Farr et al., 2017*; *Farr et al., 2019*; *Ogrodnik et al., 2019*; *Swanson et al., 2013*). As described in detail previously, SADS identification was performed on non-decalcified mouse femur sections by FISH (*Farr et al., 2017*; *Farr et al., 2019*; *Farr and Khosla, 2019*). We cut longitudinal femur sections and assessed the SADS-positive cells within the callus area. After 4 % paraformaldehyde (PFA) crosslinking of femur sections for 20 min, sections were washed three times (five minutes each in PBS), and dehydrated in ethanol (70%, 90%, 100%, each for 3 min). Sections were air dried, denatured for 10 min at 80 °C in hybridization buffer (0.1 M Tris, pH 7.2; 25 mM $MgCl_2$; 70 % formamide [Sigma-Aldrich, Saint Louis, MO], 5 % blocking reagent [Roche] with 1.0 µg/mL of Cy3-labeled (F3002) CENPB-specific [ATTCGTTGGAAACGGGA] peptide nucleic acid (PNA) probe [Panagene Inc, Korea]) and hybridized in a dark room for 2 hr at room temperature (RT). Subsequently, the femur sections were washed and mounted with vectashield DAPI-containing mounting media (Life Technologies). With a confocal microscope, SADS (i.e. decondensed and elongated centromeres) were visualized and quantified in a blinded fashion; a senescent cell was defined with a cut-off ≥4 SADS per cell (*Farr et al., 2017*; *Farr et al., 2019*).

## Western blot analysis

Protein was isolated from the pink, organic phase obtained after RNA extraction using QIAzol as previously described (*Hummon et al., 2007*). Briefly, 300 µl ethanol were added to approximately 700 µl organic phase and incubated at RT for 3 min. Upon centrifugation (2000xg, 4 °C, 5 min), the supernatant was mixed with 1.5 ml isopropanol and incubated at RT for 10 min. Samples were spun down (12,000xg, 4 °C, 10 min) and the pellet was resuspended in 2 ml 0.3 M guanidine hydrochloride in 95 % ethanol and kept at RT for 20 min. After centrifugation (12,000xg, 4 °C, 5 min), the previous step was repeated by resuspending the pellet in 0.3 M guanidine hydrochloride, incubating at RT and spinning down the sample. This step was repeated by washing the protein pellet in ethanol and air-drying at RT for 5 min. Proteins were solubilized in 0.5 ml 1 % SDS and after centrifugation (10,000xg, 4 °C, 10 min), protein concentrations were determined using a Bradford assay (Bio Rad, 5000201) according to the manufacturer's instructions. 4× Laemmli buffer (Bio Rad, 161–0747) was added, samples were boiled at 95 °C for 5 min and 20 µg protein/sample was separated by SDS-PAGE. Proteins were transferred onto a PVDF membrane (0.45 µm, IPVH00010, Millipore) and detected using antibodies targeting Cdkn2a$^{INK4A}$ (Abcam, #ab211542, 1:1000) and Beta-Actin (Cell Signaling, #4970, 1:1000). After incubation with HRP-coupled secondary antibodies, proteins were visualized using a BioRad Universal Hood II Gel Documentation System.

## Telomere-associated foci (TAF) analysis

To determine cellular senescence in the diaphyseal bone and callus, TAF was performed on murine left femora of non-decalcified methylmethacrylate (MMA)-embedded sections (n = 4/group, n = 16 altogether, all male). Our protocol was adapted from *Coppé et al., 2008* and described before (*Eckhardt et al., 2020*). In brief, bone sections were de-plasticized and hydrated in EtOH gradient followed by water and PBS. Antigen was retrieved by incubation in Tris-EDTA (pH 9.0) at 95 °C for 15 min. After cool-down and hydration with water and PBS (0.5 % Tween-20/0.1 % Triton100X), slides were placed in a blocking buffer (1:60 normal goat serum [Vector Laboratories; Cat. #S-1000] in 0.1 % BSA/PBS)

for 30 min at RT. The primary antibody targeting γ-H2A.X (Cell Signaling, #9718, 1:200) was diluted in blocking buffer and incubated overnight at 4 °C in a humid chamber. The next day, slides were washed with 0.5 % Tween-20/0.1 % Triton100X in PBS followed by PBS alone, and then incubated for 30 min with secondary goat, anti-rabbit antibody biotinylated (Vector Laboratories, #BA-1000, 1:200) in blocking buffer. Afterwards, slides were washed with 0.5 % Tween-20/0.1 % Triton100X in PBS followed by PBS alone, and then incubated for 60 min with the tertiary antibody (Cy5 Streptavidin, Vector Laboratories, #SA-1500, 1:500) in PBS. Slides were subsequently washed three times with PBS, followed by FISH for TAF detection. Following 4 % paraformaldehyde (PFA) crosslinking for 20 min, sections were washed three times (5 min each in PBS), and dehydrated in graded ice-cold EtOH (70%, 90%, and 100 % for 3 min each). Sections were dried and denatured for 10 min at 80 °C in hybridization buffer (0.1 M Tris, pH 7.2, 25 mM MgCl2, 70 % formamide [Sigma-Aldrich, Saint Louis, MO], 5 % blocking reagent [Roche] with 1.0 µg/mL of Cy-3-labeled telomere-specific [CCCTAA] peptide nucleic acid [PNA] probe [TelC-Cy3, Panagene Inc, Korea; Cat. #F1002]). The slides were then hybridized in a humid chamber for 2 hr at RT. Sections were washed and mounted with vectashield DAPI-containing mounting media (Life Technologies) prior to image acquisition and analysis. The number of TAF per cell was quantified in a blinded fashion by examining overlap of signals from the telomere probe with γ-H2A.X. The mean number of TAF per cell in bone diaphysis and/ or callus was quantified using FIJI (Image J; NIH, Bethesda, MD, USA), and the percentage (%) of TAF-positive (TAF+) cells was calculated for each mouse based on the following criteria: % of cells with ≥1 TAF, % of cells with ≥2 TAF, and % of cells with ≥3 TAF, respectively. A senescent cell was defined with a cut-off ≥3 TAF per cell (*Anderson et al., 2019*).

## Cell culture experiments

Metaphyses were cut in the right tibia, both femora and both humeri to flush the bone marrow and culture bone marrow MSCs in 75 ml flasks at 37 °C and 5 % CO2 in growth medium (DMEM, 15 % FBS, 1 % Glutamax (100 x), 1 % Anti/Anti (100 x), 0.5 % Gentamicin). The medium was changed every other day and cells were plated onto 24- or 96-well plates for functional characterization assays. Senescence was induced by treating cells with 100 µM H2O2 for 4 hr (*Dasari et al., 2006*; *Duan et al., 2005*; *Kornienko et al., 2019*; *Park et al., 2017*; *Xu et al., 2021*). In vitro senolytics treatment was performed by incubating cells with 200 nM Dasatinib and 50 µM Quercetin or DMSO as vehicle for 24 hr as described elsewhere (*Zhu et al., 2015*).

## SA-β-Gal staining

To assess senescence in vitro, cellular SA-β-Gal activity was measured as described previously (*Xu et al., 2015b*). After BMSCs were washed in PBS (pH 7.4) and fixed with 2 % formaldehyde (Sigma-Aldrich) and 25 % glutaraldehyde (Sigma-Aldrich) for 5 min, they were washed three times using PBS. Cells were then incubated in SA-β-Gal solution (1 mg/ml X-Gal, 40 mM citric acid, pH 6.0, 5 mM potassium ferrocyanide, 5 mM potassium ferricyanide, 150 mM NaCl, 2 mM MgCl$_2$) at 37 °C for 16 hr. Cells were washed in ice-cold PBS and stored in PBS at 37 °C until analysis. DAPI (Life Technologies) was used to stain nuclei for cell counting. In blinded fashion, ten images per well were taken from random fields using fluorescence microscopy (Nikon Eclipse Ti). Total cell numbers were determined automatically (IX Pico, Molecular Devices, processed using the CellReporterXpress software) and visualized using crystal violet. For this purpose, cells were fixed in 4 % formaldehyde in PBS for 20 min and, upon washing with PBS, stained with 1 % (w/v) crystal violet in 20 % ethanol for 20 min. Excess dye was removed and upon drying, images were acquired.

## Statistics

Graphical data are shown as Means ± standard error of the mean (SEM) unless otherwise specified. The sample sizes were determined based on previously conducted and published experiments (e.g. *Farr et al., 2016*; *Farr et al., 2017*) in which statistically significant differences were observed among various bone parameters in response to multiple interventions in our laboratory. The used animal numbers are indicated in the Figure Legends; all samples presented represent biological replicates. No mice, samples, or data points were excluded from analyses. Data were examined for normality and distribution using dot plots and histograms; all variables were examined for skewness and kurtosis. If the normality or equal variance assumptions for parametric analysis methods were not met, data were

analyzed using non-parametric tests (e.g. Wilcoxon Rank Sum test, Mann-Whitney U test). For parametric tests, depending on the comparison, differences between groups were analyzed by independent samples t-test or one-way ANOVA, where justified as appropriate. When ANOVA determined a statistically significant ($p < 0.05$) effect, pairwise multiple comparisons were performed and the Tukey post-hoc method was applied. Statistical analyses were performed using either GraphPad Prism (Version 9.0) or R version 4.0.2. A p-value < 0.05 (two-tailed) was considered statistically significant.

## Acknowledgements

This work was supported by the German Research Foundation (DFG, 413501650) (DS), National Institutes of Health (NIH) grants P01 AG062413 (SK, JNF), R21 AG065868 (SK, JNF), R01 DK128552 (JNF), R01 AG063707 (DGM), and Mildred Scheel postdoc fellowship by the German Cancer Aid (RLK).

## Additional information

### Funding

| Funder | Grant reference number | Author |
|---|---|---|
| National Institute on Aging | P01 AG062413 | Joshua N Farr<br>Sundeep Khosla |
| National Institute on Aging | R21 AG065868 | Joshua N Farr<br>Sundeep Khosla |
| National Institute on Aging | R01 AG063707 | David G Monroe |
| National Institute of Diabetes and Digestive and Kidney Diseases | R01 DK128552 | Joshua N Farr |
| German Research Foundation | 413501650 | Dominik Saul |

The funders had no role in study design, data collection and interpretation, or the decision to submit the work for publication.

### Author contributions

Dominik Saul, Conceptualization, Data curation, Formal analysis, Funding acquisition, Investigation, Methodology, Resources, Resources, Supervision, Supervision, Validation, Visualization, Writing – review and editing; David G Monroe, Conceptualization, Data curation, Formal analysis, Funding acquisition, Investigation, Methodology, Resources, Resources, Supervision, Validation, Visualization, Writing – review and editing; Jennifer L Rowsey, Conceptualization, Investigation, Methodology, Project administration, Writing – review and editing; Robyn Laura Kosinsky, Investigation, Methodology, Project administration, Writing – review and editing; Stephanie J Vos, Madison L Doolittle, Investigation, Methodology, Writing – review and editing; Joshua N Farr, Conceptualization, Funding acquisition, Investigation, Methodology, Project administration, Resources, Supervision, Writing – review and editing; Sundeep Khosla, Conceptualization, Data curation, Formal analysis, Funding acquisition, Investigation, Methodology, Resources, Supervision, Writing – review and editing

### Author ORCIDs

Dominik Saul http://orcid.org/0000-0002-0673-3710
Joshua N Farr http://orcid.org/0000-0002-3179-6414
Sundeep Khosla http://orcid.org/0000-0002-2936-4372

### Ethics

Animal studies were performed under protocols approved by the Institutional Animal Care and Use Committee (IACUC), and experiments were performed in accordance with Mayo Clinic IACUC guidelines.

### Decision letter and Author response

Decision letter https://doi.org/10.7554/eLife.69958.sa1

Author response https://doi.org/10.7554/eLife.69958.sa2

## Additional files

### Supplementary files
- Supplementary file 1. Mouse primer sequences.
- Transparent reporting form

### Data availability
RNA-seq data was generated from GSE152677.

The following previously published datasets were used:

| Author(s) | Year | Dataset title | Dataset URL | Database and Identifier |
|---|---|---|---|---|
| Coates BA, McKenzie JA, Buettmann EG, Liu X, Gontarz PM, Zhang B, Silva MJ | 2020 | Transcriptional profiling of intramembranous and endochondral ossification after fracture in mice | https://www.ncbi.nlm.nih.gov/geo/query/acc.cgi?acc=GSE152677 | NCBI Gene Expression Omnibus, GSE152677 |

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
