## [Decision Letter]

**Acceptance summary:**

The importance of cellular senescence in the pathogenesis of many age-related diseases is being known. However, the potential role of senescent cells in fracture healing has not been defined. In this elegant study, the authors have used publicly available mRNA-sequence data of murine femoral fractures to demonstrate increased expression of senescence and senescence-associated secretory phenotype markers during fracture healing. By using an appropriate genetic mouse model, the authors provide direct experimental evidence for the presence of senescent cells at the fracture site and that elimination of p16Ink4a expressing senescent cells led to improved fracture healing phenotype. These findings are relevant in terms of exploring the therapeutic utility of senolytic drugs to promote healing of delayed healing fractures in the elderly.

**Decision letter after peer review:**

Thank you for submitting your article "Modulation of fracture healing by the transient accumulation of senescent cells" for consideration by *eLife*. Your article has been reviewed by 3 peer reviewers, including Subburaman Mohan as the Reviewing Editor and Reviewer #1, and the evaluation has been overseen by Carlos Isales as the Senior Editor. The following individuals involved in review of your submission have agreed to reveal their identity: Roberto Pacifici (Reviewer #2); Lynda Bonewald (Reviewer #3).

Essential revisions:

1) Age and gender of mice used in various in vivo studies are not clearly described. Four month old male mice were used in the first study for the generation of closed diaphyseal femoral fractures. All NIH supported studies must include both male and female mice. It was not clear if this was this case for the studies described in this manuscript.

2) Although the callus bone volume, stiffness and maximum torque were increased in the D+Q treated mice, these changes did not reach statistical significance (Figure 6G and H). Since only 5 mice were evaluated for the fracture healing phenotype, the issue of whether lack of a significant effect is due to insufficient power should be discussed. If possible, it would be useful to increase the number of mice per group in order to achieve statistical significance. If this is not possible, the interpretation of these data should be toned down and less emphasis given to these findings in the discussion.

3) It is known that fracture healing is composed of distinct stages with changing cell populations. The authors offer the mesenchymal stem cell as the cell responsible for expressing senescence markers, but this requires further confirmation.

*Reviewer #1 (Recommendations for the authors):*

1) Four month old male mice were used in the first study for generation of closed diaphyseal femoral fractures. The age and gender of mice used for the p16Luc study as well as gender of mice the D+Q study need to be described. 2) Although the callus bone volume and stiffness were increased in the D+Q treated mice, these changes did not reach statistical significance. Since only 5 mice were evaluated for the fracture healing phenotype, the issue of whether lack of a significant effect is due to insufficient power should be discussed. 3) Figure 3. Statistical analysis show that data in panel C and G are different for the two genotypes only at P<0.05 (*). Therefore, the other designations of **, *** and *** can be eliminated. The same is true for figure 6.

*Reviewer #2 (Recommendations for the authors):*

This is a very interesting and novel study that provides much needed novel information on the role of senescent cells in fracture healing. The study was carried out using an impressive array of state of the art techniques, including analysis of human data, in vitro studies and in vivo studies in mice.

The main aims of this study were to characterize the potential appearance of senescent cells during fracture healing and establish whether targeting cellular senescence with senolytics impacts facture healing dynamics. The study provides strong evidence that senescent cells are activated during fracture repair. These cells impair fracture healing, as their pharmacological removal accelerates fracture healing.

My only concern is about the lack of statistical significance of the data presented in Figure 6G and 6H. If possible, it would be useful to increase the number of mice per group in order to achieve statistical significance. If this is not possible, the interpretation of these data should be toned down and less emphasis given to these findings in the discussion.

*Reviewer #3 (Recommendations for the authors):*

The data are very convincing with regards to the expression of senescent cells with fracture healing, however, it would be important to acknowledge and consider the different stages of fracture healing when interpreting the data-especially with regards to the assumption that it is MSCs that are responsible. There are several stages of fracture healing: days 1-5 hematoma, days 5-11 fibrocartilagenous cells, days 11-28 bony, days 18 and greater bone remodeling. Therefore, the assumption that MSCs are responsible is premature without examining the specific cell populations.

In the Abstract, it is stated that a subset of cells in the fracture callus displayed hallmarks of senescence but this subset is never characterized.

In Figure 1, it appears that p21 decreases soon after the fracture and p16 increases soon after the fracture. In the text it is stated that both are upregulated. This should be clarified.

In Figure 2, Days 4, 8, 14 and 28 post fracture are examined. It would be important to correlate with the different phases of fracture, to discuss what cell types are present as each stage has a different cell composition. For example, the initial anabolic phase is characterized by the recruitment of skeletal stem cells that go on to form the cartilaginous callus, then blood vessels, followed by bone. This anabolic stage is followed by a catabolic stage where the cartilagenous callus is replaced by primary bone and then bone remodeling occurs in which the marrow space and hematopoietic tissues are regenerated. It appears the p21 and p16 peak at day 14 when mainly cartilage is present. Could this suggest that cartilage may be the source of senescent cells? Which cells are positive for SADS and RAFs as shown in Figure 2 C and E?

Figure 3. The bioluminescent cells peaked at 14 days and were gone by 18 days. Were those chondrocytes or another cell type? As the callus volume was increased in the p16 KO, could this be due to deletion in cartilage?

Figure 3A Where is the pRT-PCR data harvested at day 28?

Figure 3B The location of the luminescence in the Day 14 image looks too high on the mouse-it appears to be on the mouse belly and not in the femur. Is this correct?

Figure 4. Again knowing what cells are present in the callus at what time could yield additional useful information. Colocalizing senescent markers with MSC markers would provide the best evidence that MSC are responsible.

In Sup. Figure 2, it is stated that the highest overlap of SASP genes was in the MSC cluster. There are two MSC clusters. Was this overlap true for both?

Figure 6. (A) is mislabeled and should be (B). Figures 6D, E, and F are the most significant data showing accelerated healing and increased callus area. There was no significant differences in mineralizaed tissue and stiffness-both bone properties. This might suggest that the major effect of D+Q was on cartilage but this did not translate into increased bone or stiffness.

Page 19, line 419 states that biomechanical bone parameters were modestly enhanced. As the data were not significant, this is not a correct statement.

There is an issue with the gender of the mice used in each experiment.

Page 21 line 460 states that both male and female animals were used but

P21 l 468 states only males were used

P21 l 480 no gender is provided for the p16 luc study

P22 l 491 D+Q, noo gender is provided.

P23 l 553 what is the sex of the 35 mice?

The gender should be made clear in the figure legend and male and female data should never be combined.

P24 l 563 Single cell RNA seq was not performed in this study. Suggest renaming the section: "Analysis of published Single Cell Rna Seq data".

P24 l 571 How was the callus dissected? Is there a published method that can be referenced? If not, the details should be provided.

P25 l 596 Where were the femur sections taken-longitudinal, cross section, location through callus, etc?

---

## [Author Response]

Essential revisions:1) Age and gender of mice used in various in vivo studies are not clearly described. Four month old male mice were used in the first study for the generation of closed diaphyseal femoral fractures. All NIH supported studies must include both male and female mice. It was not clear if this was this case for the studies described in this manuscript.

We have now clearly indicated the age and sex of the mice used in each specific experiment in the Methods section (pages 21, 22, 24 and 26) and also included this information in the Figure legends for clear transparency. Specifically:

– We have added “A-E: n=35, all female” to the legend of Figure 1.

– In the legend to Figure 2, we added “B: n=24 (n=24 in Contra, n=24 in Fx, n=6 per timepoint), all male. C-D: n=16 (n=16 in Contra, n=16 in Fx), all male. E-F: n=20 (n=4 per time point and n=4 in contra on day 14), all male.”

– In the legend of Figure 3, we added “B-C: 6 WT (3 males, 3 females), 8 *p16^Luc^* (4 males, 4 females), D-F: 6 WT (3 males, 3 females), 6 *p16^Luc^* (3 males, 3 females), G: 6 WT (3 males, 3 females), 6 *p16^Luc^* (3 males, 3 females).”

– In the legend of Figure 4, we added “A-G: n=24 (n=24 in Contra, n=24 in Fx, 6 per time point per group), all male.”.

– In the legend of Figure 5, we added “A-H: n=16, all male (n=8 in Veh, n=8 in D+Q)”.

– In the legend of Figure 6, we added “B, C: n=11 (5 Veh [3 males, 2 females], 6 DQ [3 males, 3 females]), D-G: n=34, all females (17 Veh, 17 DQ), H: n=31, all females (14 Veh, 17 DQ; note that 3 bones from the Veh group could not be analyzed due to technical issues in preparing these bones for torsional testing).”

– For B: Univariate linear model with the covariate “sex”, D-H:”

– In Suppl. Figure S1, we added: A: n=48 (n=24 in Contra, n=24 in Fx, n=6 per timepoint), all male.

– In Suppl. Figure S2, we added: A-E: n=8, all male.

– In Suppl. Figure S3, we added: A: n=11 (5 Veh [3 males, 2 females], 6 DQ [3 males, 3 females]), B: n=120 (n=60 in Wehrle [n=10 per timepoint, 5 male and 5 female], n=60 in Undale [n=10 per timepoint, 5 male and 5 female]).

Based on the availability of the mice, we designed our experiments to study both sexes, and overall the manuscript includes experiments in both male and female mice. However, for purposes of power and based on mouse availability, specific experiments had to be conducted in a single sex, and this is now explicitly stated where appropriate.

2) Although the callus bone volume, stiffness and maximum torque were increased in the D+Q treated mice, these changes did not reach statistical significance (Figure 6G and H). Since only 5 mice were evaluated for the fracture healing phenotype, the issue of whether lack of a significant effect is due to insufficient power should be discussed. If possible, it would be useful to increase the number of mice per group in order to achieve statistical significance. If this is not possible, the interpretation of these data should be toned down and less emphasis given to these findings in the discussion.

We have now substantially expanded this study and in the revision, report data on n=17 vehicle- and n=17 D+Q-treated mice following fracture in Figure 6 (in this case, all female mice were used in order to minimize variability and maximize power). We have also added additional mice to the *p16^Ink4a^-*knockout study described in Figure 3 in order to balance the sexes.

3) It is known that fracture healing is composed of distinct stages with changing cell populations. The authors offer the mesenchymal stem cell as the cell responsible for expressing senescence markers, but this requires further confirmation.

We agree, and have modified the Discussion appropriately (page 19).

Reviewer #1 (Recommendations for the authors):1) Four month old male mice were used in the first study for generation of closed diaphyseal femoral fractures. The age and gender of mice used for the p16Luc study as well as gender of mice the D+Q study need to be described.

Thank you for bringing this important issue to our attention. Please see our response in point #1 of Essential Revisions.

2) Although the callus bone volume and stiffness were increased in the D+Q treated mice, these changes did not reach statistical significance. Since only 5 mice were evaluated for the fracture healing phenotype, the issue of whether lack of a significant effect is due to insufficient power should be discussed.

We agree this is an important issue and have now substantially expanded the D+Q study. Please see our response in point #2 of Essential Revisions.

3) Figure 3. Statistical analysis show that data in panel C and G are different for the two genotypes only at P<0.05 (*). Therefore, the other designations of **, *** and *** can be eliminated. The same is true for figure 6.

Thank you – we have made these changes.

Reviewer #2 (Recommendations for the authors):This is a very interesting and novel study that provides much needed novel information on the role of senescent cells in fracture healing. The study was carried out using an impressive array of state of the art techniques, including analysis of human data, in vitro studies and in vivo studies in mice.The main aims of this study were to characterize the potential appearance of senescent cells during fracture healing and establish whether targeting cellular senescence with senolytics impacts facture healing dynamics. The study provides strong evidence that senescent cells are activated during fracture repair. These cells impair fracture healing, as their pharmacological removal accelerates fracture healing.My only concern is about the lack of statistical significance of the data presented in Figure 6G and 6H. If possible, it would be useful to increase the number of mice per group in order to achieve statistical significance. If this is not possible, the interpretation of these data should be toned down and less emphasis given to these findings in the discussion.

We agree this is an important issue and have now substantially expanded the D+Q study. Please see our response in point #2 of Essential Revisions.

Reviewer #3 (Recommendations for the authors):The data are very convincing with regards to the expression of senescent cells with fracture healing, however, it would be important to acknowledge and consider the different stages of fracture healing when interpreting the data-especially with regards to the assumption that it is MSCs that are responsible. There are several stages of fracture healing: days 1-5 hematoma, days 5-11 fibrocartilagenous cells, days 11-28 bony, days 18 and greater bone remodeling. Therefore, the assumption that MSCs are responsible is premature without examining the specific cell populations.

We agree, and this is, in fact, the focus of ongoing studies that we hope to complete in the coming year, although these data are not ready at this time for inclusion in our revision. Thus, we have appropriately modified the Discussion on page 19 in order to address this point.

In the Abstract, it is stated that a subset of cells in the fracture callus displayed hallmarks of senescence but this subset is never characterized.

We have modified the Abstract to now state: “We also identified cells in the fracture callus that displayed hallmarks of senescence, including distension of satellite heterochromatin and telomeric DNA damage; the specific identity of these cells, however, requires further characterization.”

In Figure 1, it appears that p21 decreases soon after the fracture and p16 increases soon after the fracture. In the text it is stated that both are upregulated. This should be clarified.

We have clarified the kinetics of *p21^Cip1^*and *p16^Ink4a^* changes following fracture on page 5.

In Figure 2, Days 4, 8, 14 and 28 post fracture are examined. It would be important to correlate with the different phases of fracture, to discuss what cell types are present as each stage has a different cell composition. For example, the initial anabolic phase is characterized by the recruitment of skeletal stem cells that go on to form the cartilaginous callus, then blood vessels, followed by bone. This anabolic stage is followed by a catabolic stage where the cartilagenous callus is replaced by primary bone and then bone remodeling occurs in which the marrow space and hematopoietic tissues are regenerated. It appears the p21 and p16 peak at day 14 when mainly cartilage is present. Could this suggest that cartilage may be the source of senescent cells? Which cells are positive for SADS and RAFs as shown in Figure 2 C and E?

We agree this is an important issue, and the precise identity of the transiently appearing senescent cells is the focus of ongoing work in our laboratory. We have expanded on this point in the Discussion on page 19.

Figure 3. The bioluminescent cells peaked at 14 days and were gone by 18 days. Were those chondrocytes or another cell type? As the callus volume was increased in the p16 KO, could this be due to deletion in cartilage?Figure 3A Where is the pRT-PCR data harvested at day 28?Figure 3B The location of the luminescence in the Day 14 image looks too high on the mouse-it appears to be on the mouse belly and not in the femur. Is this correct?

The question regarding what cell types develop a senescent phenotype during fracture healing is part of ongoing research in our lab, but unfortunately these data are not yet available.

Figure 3A Where is the pRT-PCR data harvested at day 28?

We apologize for the typo. We presume the Reviewer is referring to the Q=qRT-PCR on day 35 (rather than day 28); at this time point, a μCT was performed, as is shown in panel G. We changed the “qRT-PCR” into “CT” and thank the reviewer for identifying this error.

Figure 3B The location of the luminescence in the Day 14 image looks too high on the mouse-it appears to be on the mouse belly and not in the femur. Is this correct?

The reviewer is right. The unusual leg position (fully flexed knee and ankle made the signal appear too cranial when looking from the dorsal point of view. We removed the image and now just show the peak luminescence on day 16 and 18).

Figure 4. Again knowing what cells are present in the callus at what time could yield additional useful information. Colocalizing senescent markers with MSC markers would provide the best evidence that MSC are responsible.

We agree, and as noted in response to the points above, this is the focus of ongoing work in our laboratory.

In Sup. Figure 2, it is stated that the highest overlap of SASP genes was in the MSC cluster. There are two MSC clusters. Was this overlap true for both?

Interestingly, the high overlap of the selected SASP markers of Suppl. Figure 2 with one MSC cluster was only seen in MSC 2, not in MSC 1. These two MSC clusters especially differ regarding the expression of *Lepr* (MSC 2: *Lepr*^+^, MSC 1: *Lepr*^-^). We explicitly analyzed the overlap of the selected clusters and found that in comparison to the Lepr^-^ MSC cluster, the Lepr^+^ MSC cluster has a substantially higher SASP- and senescence-profile. We have now added these findings into the Suppl. Figure 2 (panel C) and also note that the subsequent analyses were done in the cluster MSC 2.

Figure 6. (A) is mislabeled and should be (B). Figures 6D, E, and F are the most significant data showing accelerated healing and increased callus area. There was no significant differences in mineralizaed tissue and stiffness-both bone properties. This might suggest that the major effect of D+Q was on cartilage but this did not translate into increased bone or stiffness.

We apologize and have corrected the labels in the Figure legend. However, in response to Essential Revisions and enhancing statistical power by adding more mice, the maximum torque and callus BV were significantly (p < 0.05) improved, indicating a more pronounced effect of D+Q, even on biomechanical bone healing properties.

Page 19, line 419 states that biomechanical bone parameters were modestly enhanced. As the data were not significant, this is not a correct statement.

With the larger samples size, we now show that the maximum torque was significantly (p < 0.05) improved, while bone stiffness increased, but this change was not statistically significant.

There is an issue with the gender of the mice used in each experiment.Page 21 line 460 states that both male and female animals were used butP21 l 468 states only males were usedP21 l 480 no gender is provided for the p16 luc studyP22 l 491 D+Q, noo gender is provided.P23 l 553 what is the sex of the 35 mice?The gender should be made clear in the figure legend and male and female data should never be combined.

As noted in our response to point #1 of Essential Revisions, we have now clearly indicated the age and sex of the mice used in each of the studies in the Methods section (page 21, 22, 24, 26) and also included this information in the Figure legends for clear transparency. Based on the availability of the mice, we aimed to study both sexes, and overall the manuscript includes experiments in both male and female mice. However, for purposes of power and based on mouse availability, specific experiments had to be conducted in a single sex, and this is now explicitly stated. In terms of combining sexes, we only combined sexes where the availability of mice was markedly limited (e.g., due to the difficulty of generating *p16^Ink4a^-*knockout mice) or where changes in gene expression were similar between sexes, and here we attempted to maintain as close a female/male distribution between the 2 groups as possible, as indicated in the Methods and Figure legends. In addition, as also now noted in the Statistical Methods section, we included sex as a co-variate in the statistical model in the limited instances when sexes were combined (please see figure legend 6: “For B: Univariate linear model with the covariate “sex”). Finally, in the critical expanded D+Q study in Figure 6D-H, we only used young adult female mice to minimize variability and maximize power, thereby specifically addressing this important issue.

P24 l 563 Single cell RNA seq was not performed in this study. Suggest renaming the section: "Analysis of published Single Cell Rna Seq data".

Agree – we have made this change.

P24 l 571 How was the callus dissected? Is there a published method that can be referenced? If not, the details should be provided.P25 l 596 Where were the femur sections taken-longitudinal, cross section, location through callus, etc?

We thank the reviewer for the opportunity to clarify these points. We have now added: “For callus analyses, callus and contralateral intact bone were removed as described in (20) and immediately homogenized in QIAzol Lysis Reagent (QIAGEN, Valencia, CA), and stored at -80°C. and “Briefly, after soft tissue removal, a 7 mm section around the fracture site (and a 7 mm section at the same location on the intact contralateral site) was extracted and homogenized in QIAzol.” Regarding the SADS analysis, we have added “We cut longitudinal femur sections and assessed the SADS-positive cells within the callus area.”